# Stochastic Controlled Averaging for Federated Learning with Communication Compression

**Xinmeng Huang**[*] **Ping Li** **Xiaoyun Li**

## Abstract

Communication compression has been an important topic in Federated Learning (FL) for alleviating the communication overhead. However, communication compression brings forth new challenges in FL due to the interplay of compression-incurred information distortion and inherent characteristics of FL such as partial participation and data heterogeneity. Despite the recent development, the existing approaches either cannot accommodate arbitrary data heterogeneity or partial participation, or require stringent conditions on compression. In this paper, we revisit the seminal stochastic controlled averaging method by proposing an equivalent but more efficient/simplified formulation with halved uplink communication costs, building upon which we propose two compressed FL algorithms, SCALLION and SCAFCOM, to support unbiased and biased compression, respectively. Both the proposed methods outperform the existing compressed FL methods in terms of communication and computation complexities. Moreover, SCALLION and SCAFCOM attain fast convergence rates under arbitrary data heterogeneity without any additional assumptions on compression errors. Experiments show that SCALLION and SCAFCOM outperform recent compressed FL methods under the same communication budget.

## 1 Introduction

Federated learning (FL) is a powerful paradigm for large-scale machine learning (Konečnỳ et al., 2016; McMahan et al., 2017; Yang et al., 2020) in situations where data and computational resources are dispersed among diverse clients such as phones, tablets, sensors, banks, hospitals (Kairouz et al., 2021). FL enjoys the advantage of distributed optimization on the efficiency of computational resources as the local clients conduct computations simultaneously. Moreover, FL provides the first layer of protection for data privacy as the local data never leaves the local device during training. Here, we first summarize the significant challenges in algorithmic development and theory of FL:

- **Severe data heterogeneity.** Unlike in classic distributed training, the local data distribution in FL can vary significantly (*i.e.*, non-iid clients), reflecting practical scenarios where local data held by clients is highly personalized (Zhao et al., 2018; Kairouz et al., 2021; Yuan et al., 2023; Li et al., 2022a). When multiple local training steps are taken, the local models become "biased" toward minimizing the local losses instead of the global loss, hindering the convergence quality of the global model (Mohri et al., 2019; Li et al., 2020c;a).

- **Partial client participation.** Not all clients can always join the training, *e.g.*, due to unstable connections or active selection (Li et al., 2020a). Consequently, only a fraction of clients are involved in each FL training round to interact with the central server. This slows down the convergence of the global model due to less accessible data/information per round (Charles et al., 2021; Chen et al., 2022; Li & Li, 2023).

- **Heavy communication workload.** The cost of model transmission can be a major challenge in FL systems with limited bandwidth (*e.g.*, portable wireless devices), especially for models with millions or billions of parameters. Therefore, communication compression, a technique that aims to reduce the volume of information transmitted, has gained growing research interests in FL (Basu et al., 2019; Reisizadeh et al., 2020; Haddadpour et al., 2021; Li & Li, 2023).

The classic FL approach, FedAvg (Konečnỳ et al., 2016; McMahan et al., 2017; Stich, 2019), performs multiple gradient-descent steps within each accessible client before communicating with the central server. FedAvg is notably hampered by data heterogeneity and partial client participation (Karimireddy et al., 2020b; Li et al., 2020c; Yang et al., 2021) due to the "client drift" effect.

---

[*]The work is conducted at LinkedIn — Bellevue, 98004 WA, USA. Xinmeng Huang is a Ph.D. student in the Graduate Group of Applied Mathematics and Computational Science at the University of Pennsylvania.

Furthermore, when communication compression is employed, the adverse effect of data heterogeneity can be amplified due to the interplay of client drift and inaccurate message aggregation caused by compression (Basu et al., 2019; Reisizadeh et al., 2020; Haddadpour et al., 2021; Mitra et al., 2021; Malekijoo et al., 2021; Li & Li, 2023); see Figure 3 in Appendix A for illustration. This naturally raises the question regarding the theory and utility of compressed FL approaches:

> *Can we design FL approaches that accommodate arbitrary data heterogeneity, local updates, and partial participation, as well as support communication compression?*

None of the existing algorithms have successfully achieved this goal in non-convex FL, despite a few studies in the strongly-convex and deterministic scenarios (Grudzień et al., 2023; Youn et al., 2022; Condat et al., 2023; Sadiev et al., 2022). For instance, FEDPAQ (Reisizadeh et al., 2020), FEDCOM (Haddadpour et al., 2021), QSPARSE-SGD (Basu et al., 2019), LOCAL-SGD-C (Gao et al., 2021) consider compressed FL algorithms under iid clients. FEDCOMGATE (Haddadpour et al., 2021), designed for unbiased compressors, does not support biased compressors, and their analysis does not consider partial client participation. FED-EF (Li & Li, 2023) focuses on biased compression with error feedback (EF) (Seide et al., 2014; Karimireddy et al., 2019) and partial client participation. However, the convergence analysis requires the assumption of bounded gradient dissimilarity on the data heterogeneity and shows an extra slow-down factor under partial participation, suggesting a theoretical limitation of EF in FL. Moreover, both Haddadpour et al. (2021) and Li & Li (2023) require assumptions on compression errors (see Appendix C.4 for more details).

## 1.1 MAIN RESULTS AND CONTRIBUTIONS

We develop SCALLION and SCAFCOM, two compressed FL algorithms for unbiased and biased compressors, respectively, that are practical to implement, robust to data heterogeneity and partial participation, and exhibit superior theoretical convergence. Table 1 presents the comparison of convergence rates of our results with prior works. Specifically, the main contributions are:

- We revisit SCAFFOLD (Karimireddy et al., 2020b) and propose a simplified formulation. The new implementation reduces the uplink communication cost by half, requiring each client to transmit only one increment variable (of the same size as the model), instead of two variables.

- Builing on the new formulation, we propose SCALLION method employs unbiased compressors for the communication of increment variables. We establish its convergence result for non-convex objectives. SCALLION obtains the state-of-the-art communication and computation complexities for FL under unbiased compressors and supports partial client participation.

- We further develop SCAFCOM which enables biased compressors for broader applications. Local momentum is applied to guarantee fast convergence and improve empirical performance. The communication and computation complexities of SCAFCOM improve prior results by significant margins, particularly when compression is aggressive.

- We conduct experiments to illustrate the effectiveness of SCALLION and SCAFCOM. Our empirical results show that the proposed methods achieve comparable performance to full-precision FL methods with substantially reduced communication costs, and outperform recent compressed FL methods under the same communication budget.

Notably, our analysis only requires the smoothness of objectives and bounded variance of stochastic gradients, *without additional assumptions on data heterogeneity or compression errors*, unlike prior works (Appendix C.4). To our best knowledge, SCALLION and SCAFCOM are the *first* stochastic FL methods that exhibit robustness to arbitrary data heterogeneity, partial participation, local updates, and accommodate communication compression relying solely on standard compressibilities.

## 2 RELATED WORK

**Communication compression & error feedback.** Two popular approaches are commonly employed to compress communication in distributed systems: quantization and sparsification, which are often modeled as unbiased or biased operators. Notable quantization examples include Sign (Seide et al., 2014; Bernstein et al., 2018), low-bit fixed rounding (Dettmers, 2016), random dithering (Alistarh et al., 2017), TernGrad (Wen et al., 2017), and natural compression (Horvóth et al., 2022). Sparsification compressors work by transmitting a small subset of entries from the input vector (Wangni et al., 2018; Stich et al., 2018). In distributed training, unbiased compressors

Table 1: The comparison of compressed FL algorithms under **full participation**. The upper half of the table is for unbiased compression, and the bottom half is for biased compression. $N$ is the number of clients; $\epsilon$ is a target for the stationarity $\mathbb{E}[\|\nabla f(\hat{x})\|^2] \leq \epsilon$; $\omega$ and $q$ are compression-related parameters (see Definitions 1 & 2). **#A. Comm.** is the total number of communication rounds when $\sigma \to 0$ asymptotically while **#A. Comp.** denotes the total number of gradient evaluations required per client when $\epsilon \to 0$ asymptotically (see Section 4.3 for details); **P.P.** denotes allowing partial client participation; **D.H.** denotes allowing arbitrary data heterogeneity; **S.C.** denotes only requiring standard compressibilities (*i.e.*, Definitions 1 & 2). Constants like objectives' smoothness $L$ and stochastic gradients' variance $\sigma^2$ are omitted for clarity.

| Algorithm | #A. Comm. | #A. Comp. | P.P. | D.H. | S.C. |
|---|---|---|---|---|---|
| FEDPAQ (Reisizadeh et al., 2020) | $\frac{1+\omega/N}{\epsilon}$ ♮ | $\frac{1+\omega}{N\epsilon^2}$ | ✔ | ✗ | ✔ |
| FEDCOM (Haddadpour et al., 2021) | $\frac{1+\omega/N}{\epsilon}$ ♮ | $\frac{1+\omega}{N\epsilon^2}$ | ✗ | ✗ | ✔ |
| FEDCOMGATE (Haddadpour et al., 2021) | $\frac{1+\omega}{\epsilon}$ | $\frac{1+\omega}{N\epsilon^2}$ | ✗ | ✔ | ✗ |
| **SCALLION (Theorem 1)** | $\frac{1+\omega}{\epsilon}$ | $\frac{1+\omega}{N\epsilon^2}$ | ✔ | ✔ | ✔ |
| **SCAFCOM[†] (Corollary 2)** | $\frac{1+\omega}{\epsilon}$ | $\frac{1}{N\epsilon^2}$ | ✔ | ✔ | ✔ |
| QSPARSE-SGD (Basu et al., 2019) | $\frac{1}{(1-q)^2\epsilon}$ | $\frac{1}{N\epsilon^2}$ | ✗ | ✗ | ✔ |
| LOCAL-SGD-C (Gao et al., 2021) | $\frac{K}{(1-q)^2\epsilon}$ | $\frac{1}{N\epsilon^2}$ | ✗ | ✗ | ✔ |
| FED-EF (Li & Li, 2023) | $\frac{1}{(1-q)^2\epsilon}$ | $\frac{1}{N(1-q)^2\epsilon^2}$ | ✔ | ✗ | ✗ |
| **SCAFCOM (Theorem 2)** | $\frac{1}{(1-q)\epsilon}$ | $\frac{1}{N\epsilon^2}$ | ✔ | ✔ | ✔ |

♮ The communication complexity requires homogeneous (iid) clients, though slightly better than ours.
[†] The results are obtained by transforming unbiased compressors into biased compressors through scaling.

usually can be applied in place of the full-precision gradients to get reasonable convergence and empirical performance. However, directly using biased compressors may slow down convergence or even lead to divergence (Beznosikov et al., 2020; Li & Li, 2023). To mitigate this, the technique of error feedback (EF) is first proposed in Seide et al. (2014), which has proven effective in addressing biased compressors (Stich et al., 2018; Karimireddy et al., 2019) and has inspired many subsequent works (Wu et al., 2018; Alistarh et al., 2018; Li et al., 2022b). Moreover, a variant scheme of EF called EF21 is introduced recently (Richtárik et al., 2021). EF21 compresses increments of deterministic gradients and offers superior theoretical guarantees compared to vanilla error feedback.

**Heterogeneity and compression in FL.** Federated learning has gained great prominence since the introduction of FEDAVG, proposed by McMahan et al. (2017) to improve the communication efficiency of classic distributed training. Subsequent studies reveal its susceptibility to data heterogeneity (*i.e.*, non-iid clients) due to the "client-drift" effect, particularly when not all clients participate in training (Stich, 2019; Yu et al., 2019b; Wang & Joshi, 2021; Lin et al., 2020; Wang et al., 2020b; Li et al., 2020c; Yang et al., 2021). Substantial efforts have been made to address client heterogeneity in FL (Liang et al., 2019; Li et al., 2020b;a; Wang et al., 2020a; Zhang et al., 2021; Haddadpour et al., 2021; Guo et al., 2023; Karimi et al., 2023; Cheng et al., 2024) and develop FL protocols involving variance reduction techniques or adaptive optimizers (Karimireddy et al., 2020b; Reddi et al., 2021; Chen et al., 2020). Notably, SCAFFOLD introduced by Karimireddy et al. (2020b) leverages control variables to mitigate the impact of data heterogeneity and partial client participation.

To further reduce communication costs, communication compression has been integrated into FL, leading to methods such as FEDPAQ (Reisizadeh et al., 2020), FEDCOMGATE (Haddadpour et al., 2021), FED-EF (Li & Li, 2023). Due to information distortion incurred by compression, those compressed FL methods either lack the robustness to data heterogeneity and partial participation, or rely on stringent conditions of compressors, surpassing standard unbiased/contractive compressibility.

**Federated learning with momentum.** The momentum mechanism in optimization traces back to Nesterov's acceleration (Yurri, 2004) and the heavy-ball method (Polyak, 1964), which have been extended to stochastic optimization (Yan et al., 2018; Yu et al., 2019a; Liu et al., 2020) and other domains (Yuan et al., 2021; He et al., 2023b;a; Chen et al., 2023; Fatkhullin et al., 2023). In the context of FL, momentum has been widely incorporated and shown to enhance performance (Wang et al., 2020b; Karimireddy et al., 2020a; Khanduri et al., 2021; Das et al., 2022; Cheng et al., 2024).

# 3 PROBLEM SETUP

Formally, in federated learning, we aim to minimize the following objective:

$$\min_{x \in \mathbb{R}^d} \quad f(x) := \frac{1}{N} \sum_{i=1}^{N} f_i(x) \quad \text{where} \quad f_i(x) := \mathbb{E}_{\xi_i \sim \mathcal{D}_i}[F(x; \xi_i)],$$

where $\xi_i$ represents a local data sample of client $i$ following data distribution $\mathcal{D}_i$, $F(x; \xi_i)$ represents the loss function evaluated at model $x$ and sample $\xi_i$, and $f_i(x)$ is the local objective w.r.t. $\mathcal{D}_i$. In practice, the data distributions $\mathcal{D}_i$ across clients may vary significantly (referred to as *data heterogeneity*), resulting in the inequality $f_i(x) \neq f_j(x)$ for different clients $i$ and $j$. Consequently, a globally stationary model $x^\star$ with $\nabla f(x^\star) = 0$ may not be a stationary point of the local objectives. In contrast, if the clients are homogeneous (*i.e.* following a common data distribution $\mathcal{D}$), we would have $f_1(x) = \cdots = f_N(x)$ and a globally stationary model would also be stationary for each client. Throughout the paper, we use $\|\cdot\|$ to denote the $\ell_2$ vector norm and use $[N]$ to denote $\{1, \ldots, N\}$ for $N \in \mathbb{N}_+$. The notation $\lesssim$ denotes inequalities that hold up to numeric numbers; $\gtrsim$ and $\asymp$ are utilized similarly. We state the following standard assumptions required for our convergence analysis.

**Assumption 1.** *Each local objective $f_i$ has $L$-Lipschitz gradient, i.e., for any $x, y \in \mathbb{R}^d$ and $1 \leq i \leq N$, it holds that $\|\nabla f_i(x) - \nabla f_i(y)\| \leq L\|x - y\|$.*

**Assumption 2.** *There exists $\sigma \geq 0$ such that for any $x \in \mathbb{R}^d$ and $1 \leq i \leq N$, $\mathbb{E}_{\xi_i}[\nabla F(x; \xi_i)] = \nabla f_i(x)$ and $\mathbb{E}_{\xi_i}[\|\nabla F(x; \xi_i) - \nabla f_i(x)\|^2] \leq \sigma^2$, where $\xi_i \sim \mathcal{D}_i$ are iid local samples at client $i$.*

## 4 SCALLION: UNBIASED COMPRESSED COMMUNICATION

In this section, we first revisit the seminal SCAFFOLD algorithm (Karimireddy et al., 2020b), which requires communicating two variables (of the same size as the model) from client to server per communication round. We present a new formulation with only a *single variable for uplink communication* for each client. We then propose SCALLION, which employs unbiased compressors to reduce the communication workload of SCAFFOLD and provide the convergence analysis.

### 4.1 BACKGROUND OF SCAFFOLD

The SCAFFOLD algorithm (Karimireddy et al., 2020b) maintains local control variables $\{c_i^t\}_{i=1}^N$ on clients and a global control variable $c^t$ on the server. Let $\mathcal{S}^t \subseteq [N]$ (with $|\mathcal{S}^t| = S$) be the set of accessible (active) clients to interact with the server in the $t$-th round. In each training round, SCAFFOLD conducts $K$ local updates within each active client $i \in \mathcal{S}^t$ by

$$y_i^{t,k+1} := y_i^{t,k} - \eta_l(\nabla F(y_i^{t,k}; \xi_i^{t,k}) - c_i^t + c^t), \quad \text{for } k = 0, \ldots, K-1, \tag{1}$$

where $y_i^{t,k}$ is the local model in client $i$ initialized with the server model $y_i^{t,0} := x^t$ and $\eta_l$ is the local learning rate. Here, the subscript $i$ represents the client index, while the superscripts $t$ and $k$ denote the outer and inner loop indexes corresponding to communication rounds and local-update steps, respectively. Upon the end of local training steps, clients update local control variables as:

$$c_i^{t+1} := \begin{cases} c_i^t - c^t + \frac{x^t - y_i^{t,K}}{\eta_l K}, & \text{if } i \in \mathcal{S}^t, \\ c_i^t, & \text{otherwise.} \end{cases} \tag{2}$$

The increments of local model $y_i^{t,K} - x^t$ and control variable $c_i^{t+1} - c_i^t$, of each participating client $i \in \mathcal{S}^t$, are then sent to the central server and aggregated to update the global model parameters:

$$x^{t+1} := x^t + \frac{\eta_g}{S} \sum_{i \in \mathcal{S}^t} (y_i^{t,K} - x^t), \quad c^{t+1} := c^t + \frac{1}{N} \sum_{i \in \mathcal{S}^t} (c_i^{t+1} - c_i^t),$$

where $\eta_g$ is the global learning rate. The detailed description of SCAFFOLD can be found in Appendix B. Notably, the control variables of SCAFFOLD track local gradients such that $c_i^t \approx \nabla f_i(x^t)$ and $c^t \approx \nabla f(x^t)$, thereby mimicking the ideal update through $\nabla F(y_i^{t,k}; \xi_i^{t,k}) - c_i^t + c^t \approx \nabla f(x^t)$ given $\nabla F(y_i^{t,k}; \xi_i^{t,k}) \approx \nabla f_i(y_i^{t,k})$ and $y_i^{t,k} \approx x^t$. Consequently, the local updates are nearly synchronized in the presence of data heterogeneity without suffering from client drift.

While the introduction of control variables enables SCAFFOLD to converge robustly with arbitrarily heterogeneous clients and partial client participation, the original implementation of SCAFFOLD described above requires clients to communicate both updates of local models $y_i^{t,K} - x^t$ and control variables $c_i^{t+1} - c_i^t$ (also see Karimireddy et al. (2020b, Alg. 1, line 13)). This results in a doubled client-to-server communication cost and more obstacles to employing communication compression, compared to its counterparts without control variables such as FEDAVG.

### 4.2 DEVELOPMENT OF SCALLION

We present an equivalent implementation of SCAFFOLD which only requires a single variable for uplink communication and is readily employable for communication compression. Expanding the

local updates $y_i^{t,K} - x^t$ and control variables $c_i^{t+1} - c_i^t$ by exploiting (1) and (2), we have

$$c_i^{t+1} - c_i^t = \frac{x^t - y_i^{t,K}}{\eta_l K} - c^t = \frac{1}{K} \sum_{k=0}^{K-1} \nabla F(y_i^{t,k}; \xi_i^{t,k}) - c_i^t \triangleq \Delta_i^t, \quad (3)$$

$$y_i^{t,K} - x^t = -\eta_l \sum_{k=0}^{K-1} \left( \nabla F(y_i^{t,k}; \xi_i^{t,k}) - c_i^t + c^t \right) = -\eta_l K (\Delta_i^t + c^t). \quad (4)$$

In (3) and (4), the updates of local models and control variables share a common component, the *increment* variables $\Delta_i^t$. Since the global control variable $c^t$ is inherently maintained by the server, updates $c_i^{t+1} - c_i^t$ and thus $y_i^{t,K} - x^t$ can be recovered by the server upon receiving $\Delta_i^t$. Hence, the global model and control variable can be equivalently updated as

$$x^{t+1} = x^t - \frac{\eta_g \eta_l K}{S} \sum_{i \in \mathcal{S}^t} \left( \Delta_i^t + c^t \right) \quad \text{and} \quad c^{t+1} = c^t + \frac{1}{N} \sum_{i \in \mathcal{S}^t} \Delta_i^t. \quad (5)$$

**We only need to communicate $\Delta_i^t$.** In the above formulation, by communicating the increment variables $\Delta_i^t$ and applying (5) at the server accordingly, SCAFFOLD can be implemented equivalently with a halved uplink communication cost, compared to the original one (Karimireddy et al., 2020b). Notably, our new implementation only modifies the communication procedure, and maintain the same local updates as in Karimireddy et al. (2020b); see Algorithm 4 in Appendix B.

**Benefits of compressing $\Delta_i^t$.** Importantly, the new implementation provides a simpler backbone for communication compression as only the transmission of $\Delta_i^t$ is to be compressed. Moreover, unlike compressing local gradients as adopted in prior literature, compressing $\Delta_i^t$ asymptotically eliminates compression errors even in the presence of data heterogeneity. Consider the case of deterministic gradients for simplicity. Based on the update rules of SCAFFOLD, if hypothetically the training approached a steady stage where $x^t$ is close to a stationary point $x^\star$, we expect to have $c_i^t \approx \nabla f_i(x^\star)$ and $c^t = \frac{1}{N} \sum_{i=1}^N c_i^t \approx \nabla f(x^\star) = 0$. Thus, the directions in local updates satisfy $\nabla f_i(y^{t,k}) - c_i^t + c^t \approx 0$ so that $x^\star \approx x^t \approx y^{t,1} \approx \cdots \approx y^{t,K}$. The definition of $\Delta_i^t$ in (3) implies $\Delta_i^t = \frac{1}{K} \sum_{k=0}^{K-1} \nabla f_i(y_i^{t,k}) - c_i^t \approx 0$. Namely, the increment variable $\Delta_i^t$ gradually vanishes as the algorithm iterates. Therefore, taking an $\omega$-unbiased compressor as an example (see Definition 1), compressing $\Delta_i^t$ results in a vanishing compression error $\mathbb{E}[\|\mathcal{C}_i(\Delta_i^t) - \Delta_i^t\|^2] \le \omega \|\Delta_i^t\|^2 \to 0$, regardless of data heterogeneity. In contrast, if one considers compressing local gradients directly, a constantly large compression error is introduced in each communication round $\mathbb{E}[\|\mathcal{C}_i(\nabla f_i(x^t)) - \nabla f_i(x^t)\|^2] \le \omega \|\nabla f_i(x^t)\|^2 \to \omega \|\nabla f_i(x^\star)\|^2 \ne 0$. The constant $\|\nabla f_i(x^\star)\|^2$ can be extremely large under severe data heterogeneity, resulting in algorithmic susceptibility to data heterogeneity.

---

**Algorithm 1** SCALLION: SCAFFOLD with single compressed uplink communication

1: **Input:** initial model $x^0$ and control variables $\{c_i^0\}_{i=1}^N$, $c^0$; local learning rate $\eta_l$; global learning rate $\eta_g$; local steps $K$; number of sampled clients $S$; scaling factor $\alpha \in [0, 1]$
2: **for** $t = 0, \cdots, T-1$ **do**
3:      Uniformly sample clients $\mathcal{S}^t \subseteq [N]$ with $|\mathcal{S}^t| = S$
4:      **for** client $i \in \mathcal{S}^t$ in parallel **do**
5:          Receive $x^t$ and $c^t$; initialize $y_i^{t,0} = x^t$
6:          **for** $k = 0, \ldots, K-1$ **do**
7:              Compute a mini-batch gradient $g_i^{t,k} = \nabla F(y_i^{t,k}; \xi_i^{t,k})$
8:              Locally update $y_i^{t,k+1} = y_i^{t,k} - \eta_l(g_i^{t,k} - c_i^t + c^t)$
9:          **end for**
10:          Compute $\delta_i^t = \alpha \left( \frac{x^t - y_i^{t,K}}{\eta_l K} - c^t \right)$
11:          Compress and send $\tilde{\delta}_i^t = \mathcal{C}_i(\delta_i^t)$ to the server      $\triangleright \alpha = 1$ and $\mathcal{C}_i = I$ recovers SCAFFOLD
12:          Update $c_i^{t+1} = c_i^t + \tilde{\delta}_i^t$ (for $i \notin \mathcal{S}^t$, $c_i^{t+1} = c_i^t$)
13:      **end for**
14:      Update $x^{t+1} = x^t - \frac{\eta_g \eta_l K}{S} \sum_{i \in \mathcal{S}^t} (\tilde{\delta}_i^t + c^t)$
15:      Update $c^{t+1} = c^t + \frac{1}{N} \sum_{i \in \mathcal{S}^t} \tilde{\delta}_i^t$
16: **end for**

---

Following the above argument regarding compression, we now propose to transmit the compressed proxy $\mathcal{C}_i(\alpha \Delta_i^t)$ of the increment variable $\Delta_i^t$, leading to the SCALLION method as presented in Algorithm 1. Here $\mathcal{C}_i$ is the compressor utilized by client $i$ while the scaling factor $\alpha \in [0,1]$ is introduced to stabilize the updates of control variables $\{c_i^t\}_{i=1}^N$ and can be viewed as the learning rate of control variables. When $\alpha = 1$ and $\{\mathcal{C}_i\}_{i=1}^N$ are the identity mappings (*i.e.*, no compression), SCALLION will reduce to SCAFFOLD with our new implementation, *i.e.*, Algorithm 4. The algorithmic comparison of SCALLION with other existing approaches is in Appendix C.2.

### 4.3 CONVERGENCE OF SCALLION

To study the convergence of SCALLION under communication compression, we first consider compressors satisfying the following standard *unbiased* compressibility.

**Definition 1** ($\omega$-UNBIASED COMPRESSOR). *There exists $\omega \geq 0$ such that for any input $x \in \mathbb{R}^d$ and each client-associated compressor $\mathcal{C}_i : \mathbb{R}^d \to \mathbb{R}^d$, $\mathbb{E}[\mathcal{C}_i(x)] = x$ and $\mathbb{E}[\|\mathcal{C}_i(x) - x\|^2] \leq \omega\|x\|^2$, where the expectation is taken over the randomness of the compressor $\mathcal{C}_i$.*

Examples that satisfy Definition 1 include random sparsification and dithering; see Appendix C.1 for details. When communication compression with $\omega$-unbiased compressors is employed, the convergence of the proposed SCALLION (Algorithm 1) is justified as follows.

**Theorem 1** (SCALLION WITH UNBIASED COMPRESSION). *Under Assumptions 1, 2, and mutually independent $\omega$-unbiased compressors, if we initialize $c_i^0 = \nabla f_i(x^0)$ and $c^0 = \nabla f(x^0)$, and set $\eta_l$, $\eta_g$, and $\alpha$ as in (29), then SCALLION converges as*

$$\frac{1}{T}\sum_{t=0}^{T-1} \mathbb{E}[\|\nabla f(x^t)\|^2] \lesssim \sqrt{\frac{(1+\omega)L\Delta\sigma^2}{SKT}} + \left(\frac{(1+\omega)N^2L^2\Delta^2\sigma^2}{S^3KT^2}\right)^{1/3} + \frac{(1+\omega)NL\Delta}{ST}, \quad (6)$$

*where $\Delta \triangleq f(x^0) - \min_x f(x)$. A detailed version and the proof are in Appendix E.*

**Asymptotic complexities of SCALLION.** When using full-batch gradients (*i.e.*, $\sigma \to 0$), all terms involving $\sigma$ in (6) vanish. Consequently, the bottleneck of FL algorithms boils down to the rounds of client-to-server communication. On the other hand, when gradients are very noisy (*i.e.*, $\sigma$ is extremely large), the $\sigma/\sqrt{T}$-dependent term dominates others in which $\sigma$ are with lower orders. In this case, the performance is mainly hampered by the number of gradient evaluations. Following (Fatkhullin et al., 2023), we refer to the total number of communication rounds in the regime $\sigma \to 0$ and gradient evaluations in the regime $\epsilon \to 0$ required by per client to attain $\mathbb{E}[\|\nabla f(\hat{x})\|^2] \leq \epsilon$ as the asymptotic communication complexity and computation complexity[1]. Theorem 1 shows $\frac{N(1+\omega)}{S\epsilon}$ asymptotic communication complexity and $\frac{1+\omega}{S\epsilon^2}$ asymptotic computation complexity of SCALLION. Here, we focus on presenting the impact of stationarity $\epsilon$, compression $\omega$, client participation $S$ and $N$, and local steps $K$ in asymptotic complexities.

**Comparison with prior compressed FL methods.** Table 1 provides a summary of non-convex FL methods employing unbiased compressors under full client participation. We observe that SCALLION matches the state-of-the-art asymptotic communication and computation complexities under non-iid clients. In particular, while having the same asymptotic complexities as FEDCOMGATE (Haddadpour et al., 2021), SCALLION does not depend on a large uniform bound of compression errors (see Appendix C.4) in convergence and thus has a superior convergence rate.

To sum up, based on the above discussion, we demonstrate that SCALLION theoretically improves existing FL methods with unbiased compression.

## 5 SCAFCOM: BIASED COMPRESSION WITH MOMENTUM

While SCALLION achieves superior convergence speed under unbiased compression, its analysis cannot be adapted to *biased* compressors (also known as *contractive* compressors) to attain fast convergence rates. In this section, we propose an algorithm called SCAFCOM as a complement of SCALLION to accommodate biased communication compression in FL.

---

[1] The complexities justify convergence rates in terms of the $\sigma/\sqrt{T}$-dependent and $1/T$-dependent terms. For instance, for the rate in (6), the asymptotic communication complexity $T \asymp \frac{N(1+\omega)}{S\epsilon}$ is derived from $\frac{(1+\omega)NL\Delta}{ST} \asymp \epsilon$ while the asymptotic computation complexity $\frac{SKT}{N} \asymp \frac{1+\omega}{N\epsilon^2}$ follows from $\sqrt{\frac{(1+\omega)L\Delta\sigma^2}{SKT}} \asymp \epsilon$.

---

**Algorithm 2** SCAFCOM: SCAFFOLD with momentum-enhanced compression

1: **Input:** initial model $x^0$ and control variables $\{c_i^0\}_{i=1}^N$, $c^0$; local learning rate $\eta_l$; global learning rate $\eta_g$; local steps $K$; number of sampled clients $S$; momentum $\beta \in [0, 1]$
2: **for** $t = 0, \cdots, T - 1$ **do**
3:     Uniformly sample clients $\mathcal{S}^t \subseteq [N]$ with $|\mathcal{S}^t| = S$
4:     **for** client $i \in \mathcal{S}^t$ in parallel **do**
5:         Receive $x^t$ and $c^t$; initialize $y_i^{t,0} = x^t$
6:         **for** $k = 0, \ldots, K - 1$ **do**
7:             Compute a mini-batch gradient $g_i^{t,k} = \nabla F(y_i^{t,k}; \xi_i^{t,k})$
8:             Locally update $y_i^{t,k+1} = y_i^{t,k} - \eta_l(g_i^{t,k} - c_i^t + c^t)$
9:         **end for**
10:         Update $v_i^{t+1} = (1 - \beta)v_i^t + \beta \left( \frac{x^t - y_i^{t,K}}{\eta_l K} + c_i^t - c^t \right)$ (for $i \notin \mathcal{S}^t$, $v_i^{t+1} = v_i^t$)
11:         Compute $\delta_i^t = v_i^{t+1} - c_i^t$
12:         Compress and send $\tilde{\delta}_i^t = \mathcal{C}_i(\delta_i^t)$ to the server       ▷ $\beta = 1$ and $\mathcal{C}_i = I$ recovers SCAFFOLD
13:         Update $c_i^{t+1} = c_i^t + \tilde{\delta}_i^t$ (for $i \notin \mathcal{S}^t$, $c_i^{t+1} = c_i^t$)
14:     **end for**
15:     Update $x^{t+1} = x^t - \frac{\eta_g \eta_l K}{S} \sum_{i \in \mathcal{S}^t} (\tilde{\delta}_i^t + c^t)$
16:     Update $c^{t+1} = c^t + \frac{1}{N} \sum_{i \in \mathcal{S}^t} \tilde{\delta}_i^t$
17: **end for**

---

### 5.1 DEVELOPMENT OF SCAFCOM

In the literature, biased compressors are commonly modeled by contractive compressibility.

**Definition 2** ($q^2$-CONTRACTIVE COMPRESSOR). *There exists $q \in [0, 1)$ such that for any input $x \in \mathbb{R}^d$ and each client-associated compressor $\mathcal{C}_i : \mathbb{R}^d \to \mathbb{R}^d$, $\mathbb{E}[\|\mathcal{C}_i(x) - x\|^2] \leq q^2\|x\|^2$, where the expectation is taken over the randomness of the compressor $\mathcal{C}_i$.*

Notably, compared to unbiased compressors satisfying Definition 1, contractive compressors, though potentially having smaller squared compression errors, *no longer enjoy the unbiasedness*; see common contractive compressors in Appendix C.1. Compared to their counterparts with unbiased compressors, approaches employing biased compressors typically (i) require stringent assumptions, *e.g.*, bounded gradients (Seide et al., 2014; Koloskova et al., 2019; Basu et al., 2019) or bounded gradient dissimilarity (Huang et al., 2022; Li & Li, 2023), (ii) rely on impractical algorithmic structure, *e.g.*, a large amount of gradient computation (Fatkhullin et al., 2021), (iii) have weak convergence guarantees, *e.g.*, worse dependence on compression parameter $q$ (Zhao et al., 2022).

Recently, Fatkhullin et al. (2023) shows that tactfully incorporating momentum into communication compression can effectively mitigate the influence of biased compression and thus presents the linear speedup in terms of the number of clients. Inspired by their findings, we introduce an extra momentum variable $v_i^t$ on each client $i$ to overcome the adverse effect of biased compression. This leads to the SCAFCOM method, as presented in Algorithm 2. When client $i$ participates in the $t$-th round, an additional momentum variable $v_i^t$ is updated as

$$v_i^{t+1} := (1 - \beta)v_i^t + \beta \left( \frac{x^t - y_i^{t,K}}{\eta_l K} + c_i^t - c^t \right) = (1 - \beta)v_i^t + \frac{\beta}{K} \sum_{k=0}^{K-1} \nabla F(y_i^{t,k}; \xi_i^{t,k}),$$

where $\{y_i^{t,k}\}$ are the intermediate local models and $\beta$ is the momentum factor. We then set $v_i^{t+1} - c_i^t = (1-\beta)v_i^t + \beta K^{-1} \sum_{k=0}^{K-1} \nabla F(y_i^{t,k}; \xi_i^{t,k}) - c_i^t$ to be (compressively) communicated, as opposed to $K^{-1} \sum_{k=0}^{K-1} \nabla F(y_i^{t,k}; \xi_i^{t,k}) - c_i^t$ in SCAFFOLD and $\alpha(K^{-1} \sum_{k=0}^{K-1} \nabla F(y_i^{t,k}; \xi_i^{t,k}) - c_i^t)$ in SCALLION. Compared to the gradient $K^{-1} \sum_{k=0}^{K-1} \nabla F(y_i^{t,k}; \xi_i^{t,k})$ yielded by a single local loop, the momentum variable $v_i^{t+1}$ has a smaller variance due to its accumulation nature, thereby refining the convergence behavior under biased compression. Finally, note that similar to SCALLION, SCAFCOM only transmits one compressed variable in the uplink communication, and recovers SCAFFOLD when $\beta = 1$ and $\{\mathcal{C}_i\}_{i=1}^N$ are the identity mapping (*i.e.*, no compression).

**Connection with SCALLION.** Notably, the difference between SCAFCOM and SCALLION lies in the utilization of momentum; see the colored highlights in Algorithm 1 and Algorithm 2.

Specifically, if we replace line 10 of SCAFCOM with the following formula:

$$v_i^{t+1} := (1-\alpha)c_i^t + \alpha\left(\frac{x^t - y_i^{t,K}}{\eta_l K} + c_i^t - c^t\right) = (1-\alpha)c_i^t + \frac{\alpha}{K}\sum_{k=0}^{K-1}\nabla F(y_i^{t,k};\xi_i^{t,k}),$$

then SCAFCOM recovers SCALLION (Algorithm 1) with $\beta = \alpha$. Note that in this case, the memorization of $v_i^t$ is *no longer needed to be retained*, which is consistent with the design of SCALLION. We also remark that the roles of the scaling factor $\alpha$ and momentum $\beta$ vary in SCAL-LION and SCAFCOM. In SCALLION, $\alpha$ stabilizes the updates of control variables $\{c_i^t\}_{i=1}^N$ while SCAFCOM sets $\beta$ to mainly address the biasedness issue of contractive compressors.

## 5.2 CONVERGENCE OF SCAFCOM

The convergence of SCAFCOM under $q^2$-contractive compression is established as follows.

**Theorem 2** (SCAFCOM WITH BIASED COMPRESSION). *Under Assumption 1, 2, and $q^2$-contractive compressors $\{\mathcal{C}_i\}_{i=1}^N$, if we initialize $c_i^0 = v_i^0 = \nabla f_i(x^0)$ and $c^0 = \nabla f(x^0)$, and set $\eta_l$, $\eta_g$, and $\beta$ as in (55), then SCAFCOM converges as*

$$\frac{1}{T}\sum_{t=0}^{T-1}\mathbb{E}[\|\nabla f(x^t)\|^2] \lesssim \sqrt{\frac{L\Delta\sigma^2}{SKT}} + \left(\frac{N^2L^2\Delta^2\sigma^2}{(1-q)S^2KT^2}\right)^{1/3} + \left(\frac{N^3L^3\Delta^3\sigma^2}{(1-q)^2S^3KT^3}\right)^{1/4} + \frac{NL\Delta}{(1-q)ST}.$$

Furthermore, it is known that one can convert any $\omega$-unbiased compressor $\mathcal{C}_i$ into a $q^2$-contractive compressors with $q^2 = \frac{\omega}{1+\omega}$ through scaling $\frac{1}{1+\omega}\mathcal{C}_i : x \mapsto \frac{1}{1+\omega}\mathcal{C}_i(x)$ (Safaryan et al., 2022, Lemma 1). Thus, SCAFCOM can also employ unbiased compressors through this scaling::

**Corollary 1** (SCAFCOM WITH UNBIASED COMPRESSION). *When employing unbiased compressors (after scaling) in communication, the convergence of SCAFCOM is upper bounded by*

$$\sqrt{\frac{L\Delta\sigma^2}{SKT}} + \left(\frac{(1+\omega)N^2L^2\Delta^2\sigma^2}{S^2KT^2}\right)^{1/3} + \left(\frac{(1+\omega)^2N^3L^3\Delta^3\sigma^2}{S^3KT^3}\right)^{1/4} + \frac{(1+\omega)NL\Delta}{ST}. \quad (7)$$

**Remark 1.** *Corollary 1 is obtained by directly plugging in the relation $q^2 = \frac{\omega}{1+\omega}$ into Theorem 2 without exploiting the unbiasedness of compressors. However, it is feasible to refine the $1/T^{2/3}$ and $1/T^{3/4}$ terms in (7) by taking advantage of unbiasedness. We omit the proof here for conciseness.*

**Asymptotic complexities of SCAFCOM.** Following the result of Theorem 2, SCAFCOM with biased compression has an asymptotic communication complexity of $\frac{N}{S(1-q)\epsilon}$ and an asymptotic computation complexity of $\frac{1}{S\epsilon^2}$ to attain $\mathbb{E}[\|\nabla f(\hat{x})\|^2] \leq \epsilon$. On the other hand, when adopting unbiased compressors with scaling, Corollary 1 reveals that SCAFCOM has an asymptotic communication complexity of $\frac{N(1+\omega)}{S\epsilon}$ and an asymptotic computation complexity of $\frac{1}{S\epsilon^2}$.

**Comparison with prior compressed FL methods.** In Table 1, we compare SCAFCOM with existing FL algorithms with biased compression under full client participation. We see that SCAFCOM outperforms prior results with biased compression in the asymptotic communication complexity by at least a factor $1/(1-q)$. Moreover, inferior to SCAFCOM, the existing FL methods with biased compression cannot tolerate unbounded data heterogeneity or even require homogeneous data. Moreover, QSPARSE-SGD (Basu et al., 2019) and LOCAL-SGD-C (Gao et al., 2021) only converge under full client participation. Notably, when employing unbiased compression, SCAFCOM enhances the asymptotic computation complexity by a factor of $1+\omega$ compared to SCALLION, surpassing all prior FL methods with unbiased compression. Furthermore, under partial client participation, our rate is better than that of FED-EF (Li & Li, 2023) by a factor of $\sqrt{N/S}$, overcoming the drawback of the standard error feedback under partial participation in federated learning.

Based on discussions in Section 5, we demonstrate that SCAFCOM, as a unified approach, outperforms existing compressed FL methods under both unbiased and biased compression.

## 6 EXPERIMENTS

We present experiments to demonstrate the efficacy of our proposed methods. Since the saving in communication costs of various compressors has been well justified in the literature (see, *e.g.*, Haddadpour et al. (2021); Li & Li (2023)), we focus on (i) validating that SCALLION and SCAF-COM can empirically match SCAFFOLD (full-precision) with substantially reduced communication costs, and (ii) showing that SCALLION and SCAFCOM outperform prior methods with the same communication budget. We defer more experimental details and results to Appendix G.

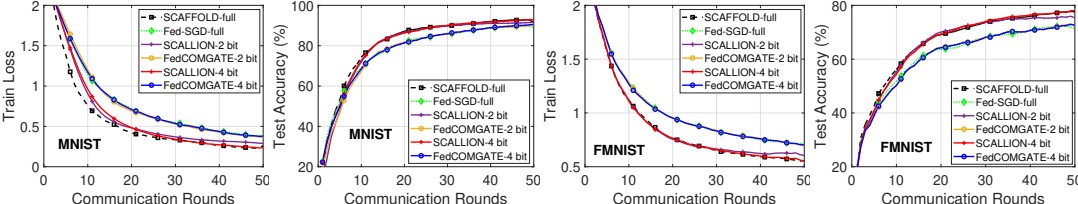

Figure 1: Train loss and test accuracy of SCAFCOM (Algorithm 2) and FED-EF (Li & Li, 2023) with biased TOP-$r$ compressors on MNIST (left half) and FMNIST (right half).

Figure 2: Train loss and test accuracy of SCALLION (Algorithm 1) and FEDCOMGATE (Haddadpour et al., 2021) with unbiased random dithering on MNIST (left half) and FMNIST (right half).

We compared the proposed algorithms with baselines including FED-EF (Li & Li, 2023), FED-COMGATE (Haddadpour et al., 2021), FED-SGD (also known as FEDAVG (Yang et al., 2021)), and SCAFFOLD (Karimireddy et al., 2020b) on two standard FL datasets: MNIST (LeCun, 1998), FMNIST (Xiao et al., 2017)). Note that FED-SGD and SCAFFOLD employ full-precision (*i.e.*, uncompressed) communication and we implement SCAFFOLD with our new formulation (Algorithm 4) for a fair comparison. We simulate biased compression with TOP-$r$ operators (see Example 3). Specifically, we adopt TOP-0.01 and TOP-0.05, where only the largest $1\%$ and $5\%$ entries in absolute values are transmitted in communication. For unbiased compression, we utilize random dithering (see Example 2), with 2 bits and 4 bits per entry, respectively.

Since all the compressed FL methods in our experiments transmit one variable in the uplink communication, their communication costs are essentially the same when the same compressor is applied. Therefore, for clarity of comparisons, we will plot the metrics versus the number of training rounds.

**SCAFCOM with biased compression.** In Figure 1, we present the train loss and test accuracy of our proposed SCAFCOM ($\beta = 0.2$) and FED-EF (Li & Li, 2023), both using biased TOP-$r$ compressors. We observe on both datasets: (i) under the same degree of compression (*i.e.*, the value of $r$ here), SCAFCOM outperforms FED-EF in terms of both training loss and test accuracy, thanks to controlled variables and the local momentum in SCAFCOM; (ii) SCAFCOM with TOP-0.01 can achieve very close test accuracy as SCAFFOLD (full-precision), and SCAFCOM with TOP-0.05 match those of SCAFFOLD, leading to the same performance while saving 20 - 100x uplink communication costs; (iii) the performance of SCAFCOM and FED-EF approaches that of the full-precision counterparts (*i.e.*, SCAFFOLD and FED-SGD) as compression assuages.

**SCALLION with unbiased compression.** In Figure 2, we plot the same set of experimental results and compare SCALLION ($\alpha = 0.1$) with FedCOMGATE (Haddadpour et al., 2021), both applying unbiased random dithering (Alistarh et al., 2017) with 2 and 4 bits per entry. Similarly, we see that SCALLION outperforms FedCOMGATE under the same compressors. The SCALLION curves of both 2-bit and 4-bit compression basically overlap those of SCAFFOLD, and 4-bit compression slightly performs better than 2-bit compression in later training rounds. Since random dithering also introduces sparsity in compressed output, the 4-bit compressor already provides around 100x communication compression, and the 2-bit compressor saves even more communication costs.

## 7 CONCLUSION

This paper proposes two compressed federated learning (FL) algorithms, SCALLION and SCAF-COM, to support unbiased and biased compression in FL. The proposed methods are built upon our new implementation of stochastic controlled averaging, along with local momentum, and communication compression. Theoretically, under minimal assumptions, SCALLION and SCAFCOM match or improve the state-of-the-art convergence of compressed FL methods. Moreover, SCAL-LION and SCAFCOM are the *first* stochastic FL methods, to the best of our knowledge, that exhibit robustness to arbitrary data heterogeneity, partial participation, local updates, and also accommodate communication compression relying solely on standard compressibilities. Empirically, experiments show that SCALLION and SCAFCOM outperform prior compressed FL methods and perform comparably to full-precision FL approaches at a substantially reduced communication cost.

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

## A INTERPLAY OF CLIENT-DRIFT AND COMMUNICATION COMPRESSION

We depict the evolution of local and global models in a client-server communication round of compressed FEDAVG with 2 clients with 3 local steps as Figure 3.

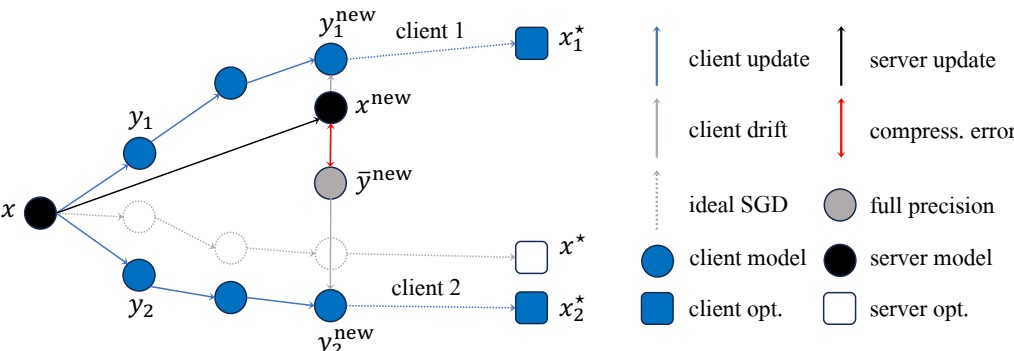

Figure 3: Interplay of client-drift and inaccurate message aggregation incurred by communication compression in FEDAVG is illustrated for 2 clients with 3 local steps (*i.e.*, $S = N = 2$, $K = 3$). The client updates $y_i$ (blue circle) move towards the individual client optima $x_i^\star$ (blue square). The server updates (black circle) move towards a *distorted proxy*, depending on the degree of compression, of the full-precision averaged model $\frac{1}{N} \sum_{i=1}^{N} x_i^\star$ (grey circle), instead of the true optimum $x^\star$ (white square).

## B DETAILED IMPLEMENTATIONS OF SCAFFOLD

---

**Algorithm 3** SCAFFOLD: Stochastic controlled averaging for FL (Karimireddy et al., 2020b)

---

1: **Input:** initial model $x^0$ and control variables $\{c_i^0\}_{i=1}^N$, $c^0$; local learning rate $\eta_l$; global learning rate $\eta_g$; local steps $K$; number of sampled clients $S$
2: **for** $t = 0, \cdots, T - 1$ **do**
3:     Uniformly sample clients $\mathcal{S}^t \subseteq [N]$ with $|\mathcal{S}^t| = S$
4:     **for** client $i \in \mathcal{S}^t$ in parallel **do**
5:         Receive $x^t$ and $c^t$; initialize $y_i^{t,0} = x^t$
6:         **for** $k = 0, \ldots, K - 1$ **do**
7:             Compute a mini-batch gradient $g_i^{t,k} = \nabla F(y_i^{t,k}; \xi_i^{t,k})$
8:             Locally update $y_i^{t,k+1} = y_i^{t,k} - \eta_l(g_i^{t,k} - c_i^t + c^t)$
9:         **end for**
10:         Update $c_i^{t+1} = c_i^t - c^t + \frac{x^t - y_i^{t,K}}{\eta_l K}$ (for $i \notin \mathcal{S}^t$, $c_i^{t+1} = c_i^t$)
11:         Send $y_i^{t,K} - x^t$ and $c_i^{t+1} - c_i^t$ to the server
12:     **end for**
13:     Update $x^{t+1} = x^t + \frac{\eta_g}{S} \sum_{i \in \mathcal{S}^t} (y_i^{t,K} - x^t)$
14:     Update $c^{t+1} = c^t + \frac{1}{N} \sum_{i \in \mathcal{S}^t} (c_i^{t+1} - c_i^t)$
15: **end for**

---

The original implementation of SCAFFOLD (Karimireddy et al., 2020b) is stated in Algorithm 3 where no compression is employed in communication. In this implementation, each participating client needs to transmit the increments of both local model $y^{t,K} - y^{t,0}$ and control variable $c_i^{t+1} - c_i^t$ to the server at the end of local updates, resulting to two rounds of uplink communication for per training iteration.

By communicating the increment variable $\Delta_i^t$, we can implement SCAFFOLD equivalently with only a single round of uplink communication for each participating client, as described in Algorithm 4.

---

**Algorithm 4** A simplified formulation of SCAFFOLD with single-variable uplink communication

---

1: **Input:** initial model $x^0$ and control variables $\{c_i^0\}_{i=1}^N$, $c^0$; local learning rate $\eta_l$; global learning rate $\eta_g$; local steps $K$; number of sampled clients $S$
2: **for** $t = 0, \cdots, T-1$ **do**
3:     Uniformly sample clients $\mathcal{S}^t \subseteq [N]$ with $|\mathcal{S}^t| = S$
4:     **for** client $i \in \mathcal{S}^t$ in parallel **do**
5:         Receive $x^t$ and $c^t$; initialize $y_i^{t,0} = x^t$
6:         **for** $k = 0, \ldots, K-1$ **do**
7:             Compute a mini-batch gradient $g_i^{t,k} = \nabla F(y_i^{t,k}; \xi_i^{t,k})$
8:             Locally update $y_i^{t,k+1} = y_i^{t,k} - \eta_l(g_i^{t,k} - c_i^t + c^t)$
9:         **end for**
10:         Compute $\Delta_i^t = \frac{x^t - y_i^{t,K}}{\eta_l K} - c^t$ and send $\Delta_i^t$ to the server
11:         Update $c_i^{t+1} = c_i^t + \Delta_i^t$ (for $i \notin \mathcal{S}^t$, $c_i^{t+1} = c_i^t$)
12:     **end for**
13:     Update $x^{t+1} = x^t - \frac{\eta_g \eta_l K}{S} \sum_{i \in \mathcal{S}^t}(\Delta_i^t + c^t)$
14:     Update $c^{t+1} = c^t + \frac{1}{N} \sum_{i \in \mathcal{S}^t} \Delta_i^t$
15: **end for**

---

## C   MORE DETAILS, DISCUSSIONS, AND COMPARISONS

### C.1   UNBIASED AND BIASED COMPRESSORS

Examples of popular unbiased compressors include:

**Example 1** (RANDOM SPARSIFICATION (WANGNI ET AL., 2018)). *For any $s \in [d]$, the random-s sparsification is defined as $\mathcal{C} : x \mapsto \frac{d}{s}(\xi \odot x)$ where $\odot$ denotes the entry-wise product and $\xi \in \{0,1\}^d$ is a uniformly random binary vector with $s$ non-zero entries. This random-s sparsification is an $\omega$-unbiased compressor with $\omega = d/s - 1$.*

**Example 2** (RANDOM DITHERING (ALISTARH ET AL., 2017)). *For any $b \in \mathbb{N}_+$, the random dithering with b-bits per entry is defined as $\mathcal{C} : x \mapsto \|x\| \times \text{sign}(x) \odot \zeta(x)$ where $\{\zeta_k\}_{k=1}^d$ are independent random variables such that*

$$\zeta_k(x) := \begin{cases} \lfloor 2^b |x_k|/\|x\| \rfloor /2^b, & \text{with probability } \lceil 2^b |x_k|/\|x\| \rceil - 2^b |x_k|/\|x\|, \\ \lceil 2^b |x_k|/\|x\| \rceil /2^b, & \text{otherwise,} \end{cases}$$

*where $\lfloor \cdot \rfloor$ and $\lceil \cdot \rceil$ are the floor and ceiling functions, respectively. This random dithering with b-bits per entry is an $\omega$-unbiased compressor with $\omega = \min\{d/4^b, \sqrt{d}/2^b\}$.*

Common biased compressors include (Li & Li, 2023):

**Example 3** (TOP-$r$ OPERATOR). *For any $r \in [0,1]$, the Top-r operator is defined as $\mathcal{C} : x \mapsto (\mathbb{1}\{k \in \mathcal{S}_r(x)\} x_k)_{k=1}^d$ where $\mathcal{S}_r(x)$ is the set of the largest $r \times d$ entries of $x$ in absolute values. Top-r operator is a $q^2$-contractive compressor with $q^2 = 1 - r$.*

**Example 4** (GROUPED SIGN). *Given a partition of $[d]$ with $M$ groups (e.g., layers of neural networks) $\{\mathcal{I}_m\}_{m=1}^M$, the grouped sign with partition $\{\mathcal{I}_m\}_{m=1}^M$ is defined as $\mathcal{C} : x \mapsto \sum_{m=1}^M \|x_{\mathcal{I}_m}\|_1 \odot \text{sign}(x)_{\mathcal{I}_m}/|\mathcal{I}_m|$. This grouped sign operator is a $q^2$-contractive compressor with $q^2 = 1 - 1/\max_{1 \leq m \leq M} |\mathcal{I}_m|$.*

### C.2   COMPARISON OF SCALLION WITH PRIOR METHODS

**Comparison with** FEDPAQ (Reisizadeh et al., 2020)**,** FEDCOM (Haddadpour et al., 2021)**,** FED-EF (Li & Li, 2023)**.** All of them boil down to the FEDAVG algorithm (McMahan et al., 2017) when no compression is conducted. As such, their convergence is significantly hampered by data heterogeneity across clients due to client drift. The former two works do not consider partial participation, and FED-EF suffers from an extra slow-down factor in the convergence rate under partial participation. In opposition, SCALLION roots from SCAFFOLD, and is robust to arbitrary data heterogeneity and partial participation.

**Comparison with** FEDCOMGATE **(Haddadpour et al., 2021).** FEDCOMGATE applies compression over the uplink communication of the VRL-SGD algorithm (Liang et al., 2019), a gradient-tracking-based FL method which is different from SCAFFOLD. FEDCOMGATE suggests conducting $K = O(1/(N\epsilon))$ local steps, demanding solving local problems to an extremely accurate resolution. Moreover, it additionally requires uniformly bounded compression errors $\mathbb{E}[\|\frac{1}{N}\sum_{i\in[N]}\mathcal{C}_i(x_i)\|^2 - \|\frac{1}{N}\sum_{i\in[N]}x_i\|^2] \leq G_A^2$, which are invalid for practical compressors such as random sparsification (Wangni et al., 2018) and random dithering (Alistarh et al., 2017). In addition, both the convergence for FEDCOMGATE and VRL-SGD is only established when all clients participate in training. It is unclear if their convergence can be adapted to partial client participation. In contrast, SCALLION converges at a state-of-the-art rate that admits a flexible number of local steps and client sampling and employs standard unbiased compressors (*i.e.*, Definition 1).

## C.3 CONNECTION BETWEEN SCAFCOM AND ERROR FEEDBACK

SCAFCOM does not directly pertain to the vanilla error feedback (Seide et al., 2014; Stich, 2019), a technique widely used to tackle biased compression, but relates to the newly proposed EF21 mechanism (Richtárik et al., 2021). If one sets $\beta = 1$ in SCAFCOM, then the message $K^{-1}\sum_{k=0}^{K-1}\nabla F(y_i^{t,k};\xi_i^{t,k}) - c_i^t$ would be compressed and the control variable would be updated as $c_i^{t+1} = c_i^t + \mathcal{C}_i(K^{-1}\sum_{k=0}^{K-1}\nabla F(y_i^{t,k};\xi_i^{t,k}) - c_i^t)$. Under the simplification where $\sigma = 0$ (*i.e.*, full-batch gradients), $K = 1$ (*i.e.*, no local updates), $S = N$ (*i.e.*, full-client participation), it becomes $c_i^{t+1} = c_i^t + \mathcal{C}_i(\nabla f_i(x^t) - c_i^t)$ and the global model is updated as $x^{t+1} = x^t - \eta_g\eta_l c^{t+1}$ with $c^{t+1} = \frac{1}{N}\sum_{i=1}^N c_i^{t+1}$, recovering the recursion of EF21.

## C.4 WEAK ASSUMPTIONS OF SCALLION AND SCAFCOM

Due to comprehensive challenges in compressed FL, to facilitate convergence analysis, most existing approaches require additional stringent conditions or assumptions that are not necessarily valid in practice, including but not restricted to:

$$\max_{1\leq i\leq N}\|\nabla f_i(x)\| \leq G, \qquad \text{(Bounded Gradient Norm (Basu et al., 2019))}$$

$$\frac{1}{N}\sum_{i=1}^N \|\nabla f_i(x) - \nabla f(x)\|^2 \leq \zeta^2,$$

(Bounded Gradient Dissimilarity (Jiang & Agrawal, 2018; Li & Li, 2023; Gao et al., 2021))

$$\mathbb{E}\left[\left\|\frac{1}{N}\sum_{i=1}^N \mathcal{C}_i(x_i)\right\|^2 - \left\|\frac{1}{N}\sum_{i=1}^N x_i\right\|^2\right] \leq G_A^2.$$

(Bounded Compression Error (Haddadpour et al., 2021))

$$\mathbb{E}\left[\left\|\frac{1}{N}\sum_{i=1}^N \mathcal{C}_i(x_i) - \frac{1}{N}\sum_{i=1}^N x_i\right\|^2\right] \leq q_A^2 \left\|\frac{1}{N}\sum_{i=1}^N x_i\right\|^2.$$

(Averaged Contraction (Alistarh et al., 2018; Li & Li, 2023))

As a result, their convergence rates inevitably depend on the large constants $G, \zeta, G_A, (1-q_A)^{-1}$. In contrast, the convergence results of SCALLION and SCAFCOM presented do **not** rely on any such condition.

# D PRELIMINARIES OF PROOFS

Letting $\gamma \triangleq \eta_g \eta_l K$, the recursion of SCALLION and SCAFCOM can be formulated as $x^{t+1} = x^t - \gamma \tilde{d}^{t+1}$ where $\tilde{d}^{t+1} = \frac{1}{S} \sum_{i \in \mathcal{S}^t} \tilde{\delta}_i^t + c^t$ and $\tilde{\delta}_i^t = \mathcal{C}_i(\delta_i^t)$. For clarity, we let $g_i^t \triangleq \frac{1}{K} \sum_{k=0}^{K-1} g_i^{t,k}$, $g^t \triangleq \frac{1}{N} \sum_{i=1}^{N} g_i^t$, $d^{t+1} \triangleq \frac{1}{S} \sum_{i \in \mathcal{S}^t} \delta_i^t + c^t$. We will abbreviate $\sum_{i=1}^{N}, \sum_{k=0}^{K-1}, \sum_{i=1}^{N} \sum_{k=0}^{K-1}$ as $\sum_i$, $\sum_k$, $\sum_{i,k}$, respectively, when there is no confusion. We also define the auxiliary variable $U^t := \frac{1}{NK} \sum_{i=1}^{N} \sum_{k=1}^{K-1} \mathbb{E}[\|y_i^{t,k} - x^t\|]^2$ to facilitate the analyses. It is worth noting that $c^t \equiv \frac{1}{N} \sum_{i=1}^{N} c_i^t$ for all $t \geq 0$ in both SCALLION and SCAFCOM. Besides, the exclusive recursions are as follows.

For SCALLION. Due to client sampling, it holds that

$$c_i^{t+1} = \begin{cases} c_i^t + \mathcal{C}_i(\alpha(g_i^t - c_i^t)) & \text{if } i \in \mathcal{S}^t \\ c_i^t & \text{otherwise} \end{cases} \tag{8}$$

and $d^{t+1} = \frac{1}{S} \sum_{i \in \mathcal{S}^t} \alpha(g_i^t - c_i^t) + c^t$

For SCAFCOM. We additionally let $u_i^{t+1} \triangleq v_i^t + \beta(g_i^t - v_i^t)$. Then, due to client sampling, it holds that

$$(v_i^{t+1}, c_i^{t+1}) = \begin{cases} (u_i^{t+1}, c_i^t + \mathcal{C}_i(u_i^{t+1} - c_i^t)) & \text{if } i \in \mathcal{S}^t \\ (v_i^t, c_i^t) & \text{otherwise} \end{cases} \tag{9}$$

and $d^{t+1} = \frac{1}{S} \sum_{i \in \mathcal{S}^t} (u_i^{t+1} - c_i^t) + c^t$. Similarly, we let $v^t \triangleq \frac{1}{N} \sum_{i=1}^{N} v_i^t$ and $u^{t+1} \triangleq \frac{1}{N} \sum_{i=1}^{N} u_i^{t+1} = (1 - \beta)v^t + \beta g^t$.

Let $\mathcal{F}^{-1} = \emptyset$ and $\mathcal{F}_i^{t,k} := \sigma(\cup\{\xi_i^{t,j}\}_{0 \leq j < k} \cup \mathcal{F}^{t-1})$ and $\mathcal{F}^t := \sigma((\cup_{i \in [N]} \mathcal{F}_i^{t,K}) \cup \{\mathcal{S}^t\})$ for all $t \geq 0$ where $\sigma(\cdot)$ means the $\sigma$-algebra. We use $\mathbb{E}[\cdot]$ to indicate the expectation taking all sources of randomness into account. We will frequently use the following fundamental lemmas.

**Lemma 1** ((Cheng et al., 2024))**.** *Given any $a_1, \cdots, a_N, b \in \mathbb{R}^d$ and $a = \frac{1}{N} \sum_{i \in [N]} a_i$ and uniform sampling $\mathcal{S} \subset [N]$ without replacement such that $|\mathcal{S}| = S$, it holds that*

$$\mathbb{E}_{\mathcal{S}} \left[ \left\| \frac{1}{S} \sum_{i \in \mathcal{S}} a_i \right\|^2 \right] \leq \|a\|^2 + \frac{1}{SN} \sum_i \|a_i - a\|^2 \leq \|a\|^2 + \frac{1}{SN} \sum_i \|a_i - b\|^2.$$

**Lemma 2** ((Karimireddy et al., 2020b))**.** *Suppose $\{X_1, \cdots, X_\tau\} \subseteq \mathbb{R}^d$ be random variables that are potentially dependent. If their marginal means and variances satisfy $\mathbb{E}[X_i] = \mu_i$ and $\mathbb{E}[\|X_i - \mu_i\|^2] \leq \sigma^2$, then it holds that*

$$\mathbb{E} \left[ \left\| \sum_{i=1}^{\tau} X_i \right\|^2 \right] \leq \left\| \sum_{i=1}^{\tau} \mu_i \right\|^2 + \tau^2 \sigma^2.$$

*If they are correlated in the Markov sense with conditional means and variances satisfying $\mathbb{E}[X_i | X_{i-1}, \cdots X_1] = \mu_i$ and $\mathbb{E}[\|X_i - \mu_i\|^2] \leq \sigma^2$ [XH: check here $\mathbb{E}[\|X_i - \mu_i\|^2 \mid \mu_i] \leq \sigma^2$]. Then a tighter bound holds*

$$\mathbb{E} \left[ \left\| \sum_{i=1}^{\tau} X_i \right\|^2 \right] \leq 2\mathbb{E} \left[ \left\| \sum_{i=1}^{\tau} \mu_i \right\|^2 \right] + 2\tau \sigma^2.$$

**Lemma 3.** *Under Assumption 1, for any $\theta \in [0, 1]$ and $v, v_1, \ldots, v_N \in \mathcal{F}^{t-1}$, it holds that*

$$\mathbb{E}[\|(1 - \theta)v + \theta(g^t - \nabla f(x^t))\|^2]$$

$$\leq \min \left\{ 2\mathbb{E}[\|(1 - \theta)v\|^2] + 3\theta^2 L^2 U^t, (1 - \theta)\mathbb{E}[\|v\|^2] + 2\theta L^2 U^t \right\} + \frac{2\theta^2 \sigma^2}{NK} \tag{10}$$

*and*

$$\frac{1}{N} \sum_i \mathbb{E}[\|(1 - \theta)v_i + \theta(g_i^t - \nabla f_i(x^t))\|^2]$$

$$\leq \min \left\{ \frac{2}{N} \sum_i \mathbb{E}[\|(1 - \theta)v_i\|^2] + 3\theta^2 L^2 U^t, \frac{1 - \theta}{N} \sum_i \mathbb{E}[\|v_i\|^2] + 2\theta L^2 U^t \right\} + \frac{2\theta^2 \sigma^2}{K}. \tag{11}$$

*Proof.* We mainly focus on proving (10) as (11) can be established similarly. Using Lemma 2, we have

$$\mathbb{E}[\|(1-\theta)v + \theta(g^t - \nabla f(x^t))\|^2]$$

$$=\mathbb{E}[\|(1-\theta)v\|^2] + \mathbb{E}\left[\left\langle (1-\theta)v, \frac{\theta}{NK}\sum_{i,k}\nabla F(y_i^{t,k};\xi_i^{t,k}) - \nabla f_i(x^t)\right\rangle\right]$$

$$+\theta^2\mathbb{E}\left[\left\|\frac{1}{NK}\sum_{i,k}\nabla F(y_i^{t,k};\xi_i^{t,k}) - \nabla f_i(x^t)\right\|^2\right]$$

$$\leq\mathbb{E}[\|(1-\theta)v\|^2] + \mathbb{E}\left[\left\langle (1-\theta)v, \frac{\theta}{NK}\sum_{i,k}\nabla f_i(y_i^{t,k}) - \nabla f_i(x^t)\right\rangle\right]$$

$$+2\theta^2\mathbb{E}\left[\left\|\frac{1}{NK}\sum_{i,k}\nabla f_i(y_i^{t,k}) - \nabla f_i(x^t)\right\|^2\right] + \frac{2\theta^2\sigma^2}{NK}$$

$$\leq\mathbb{E}\left[\left\|(1-\theta)v + \frac{\theta}{NK}\sum_{i,k}\nabla f_i(y_i^{t,k}) - \nabla f_i(x^t)\right\|^2\right]$$

$$+\theta^2\mathbb{E}\left[\left\|\frac{1}{NK}\sum_{i,k}\nabla f_i(y_i^{t,k}) - \nabla f_i(x^t)\right\|^2\right] + \frac{2\theta^2\sigma^2}{NK}.$$

By further applying Sedrakyan's inequality $\|(1-\theta)v + \theta v'\|^2 \leq (1-\theta)\|v\|^2 + \theta\|v'\|^2$ and Assumption 1, we have

$$\mathbb{E}[\|(1-\theta)v + \theta(g^t - \nabla f(x^t))\|^2]$$

$$\leq(1-\theta)\mathbb{E}[\|v\|^2] + \theta(1+\theta)\mathbb{E}\left[\left\|\frac{1}{NK}\sum_{i,k}\nabla f_i(y_i^{t,k}) - \nabla f_i(x^t)\right\|^2\right] + \frac{2\theta^2\sigma^2}{NK}$$

$$\leq(1-\theta)\mathbb{E}[\|v\|^2] + 2\theta L^2 U^t + \frac{2\theta^2\sigma^2}{NK}.$$

The other upper bound of (10) follows from $\|(1-\theta)v + \theta v'\|^2 \leq 2\|(1-\theta)v\|^2 + 2\theta^2\|v'\|^2$. $\quad\square$

# E  PROOF OF SCALLION

In this section, we prove the convergence result of SCALLION with unbiased compression, where we additionally define $x^{-1} := x^0$. We thus have $\mathbb{E}[\|x^t - x^{t-1}\|^2] = 0$ for $t = 0$. Note that $x^{-1}$ is defined for the purpose of notation and is not utilized in our algorithms.

**Lemma 4** (DESCENT LEMMA). *Under Assumptions 1 and 2, it holds for all $t \geq 0$ and $\gamma > 0$ that*

$$\mathbb{E}[f(x^{t+1})]$$

$$\leq \mathbb{E}[f(x^t)] - \frac{\gamma}{2}\mathbb{E}[\|\nabla f(x^t)\|^2]$$

$$- \left(\frac{1}{2\gamma} - \frac{L}{2}\right)\mathbb{E}[\|x^{t+1} - x^t\|^2] + 4\gamma\left(1 + \frac{(1+\omega)\alpha^2}{S}\right)L^2\mathbb{E}[\|x^t - x^{t-1}\|^2]$$

$$+ \frac{8\gamma(1+\omega)\alpha^2\sigma^2}{SK} + 4\gamma\left(1 + \frac{(1+\omega)\alpha^2}{S}\right)L^2 U^t$$

$$+ \gamma\left(4\mathbb{E}[\|c^t - \nabla f(x^{t-1})\|^2] + \frac{4(1+\omega)\alpha^2}{S}\frac{1}{N}\sum_i \mathbb{E}[\|c_i^t - \nabla f_i(x^{t-1})\|^2]\right). \tag{12}$$

*Proof.* By Lemma 2 of Li et al. (2021), we have

$$f(x^{t+1}) \leq f(x^t) - \frac{\gamma}{2}\|\nabla f(x^t)\|^2 - \left(\frac{1}{2\gamma} - \frac{L}{2}\right)\|x^{t+1} - x^t\|^2 + \frac{\gamma}{2}\|\tilde{d}^{t+1} - \nabla f(x^t)\|^2$$

where $\tilde{d}^{t+1} = \frac{1}{S}\sum_{i\in\mathcal{S}^t}\tilde{\delta}_i^t + c^t$. Letting $d^{t+1} \triangleq \frac{1}{S}\sum_{i\in\mathcal{S}^t}\delta_i^t + c^t$, we further have

$$f(x^{t+1}) \leq f(x^t) - \frac{\gamma}{2}\|\nabla f(x^t)\|^2 - \left(\frac{1}{2\gamma} - \frac{L}{2}\right)\|x^{t+1} - x^t\|^2$$

$$+ \gamma\|\tilde{d}^{t+1} - d^{t+1}\|^2 + \gamma\|d^{t+1} - \nabla f(x^t)\|^2. \tag{13}$$

For $\|d^{t+1} - \nabla f(x^t)\|^2$, using Lemma 2 and the fact that $c^t \equiv \frac{1}{N}\sum_i c_i^t$ and $d^{t+1} = \frac{1}{S}\sum_{i\in\mathcal{S}^t}\alpha(g_i^t - c_i^t) + c^t$, we have

$$\mathbb{E}[\|d^{t+1} - \nabla f(x^t)\|^2]$$

$$= \mathbb{E}\left[\left\|c^t + \frac{1}{S}\sum_{i\in\mathcal{S}^t}\alpha(g_i^t - c_i^t) - \nabla f(x^t)\right\|^2\right]$$

$$\leq \mathbb{E}[\|(1-\alpha)(c^t - \nabla f(x^t)) + \alpha(g^t - \nabla f(x^t))\|^2] + \frac{\alpha^2}{SN}\sum_i\mathbb{E}[\|g_i^t - c_i^t\|^2]. \tag{14}$$

Using (10) and Assumption 1, we further have

$$\mathbb{E}[\|(1-\alpha)(c^t - \nabla f(x^t)) + \alpha(g^t - \nabla f(x^t)\|^2]$$

$$\leq 2\mathbb{E}[\|c^t - \nabla f(x^t)\|^2] + 3\alpha^2 U^t + \frac{2\alpha^2\sigma^2}{NK}$$

$$\leq 4\mathbb{E}[\|c^t - \nabla f(x^{t-1})\|^2] + 4L^2\mathbb{E}[\|x^t - x^{t-1}\|^2] + 3\alpha^2 L^2 U^t + \frac{2\alpha^2\sigma^2}{NK}. \tag{15}$$

Similarly, using (11) and Assumption 1, we have

$$\frac{\alpha^2}{SN}\sum_i\mathbb{E}[\|g_i^t - c_i^t\|^2] \leq \frac{2\alpha^2}{SN}\sum_i\mathbb{E}[\|c_i^t - \nabla f_i(x^t)\|^2] + \frac{2\alpha^2}{SN}\sum_i\mathbb{E}[\|g_i^t - \nabla f_i(x^t)\|^2]$$

$$\leq \frac{2\alpha^2}{SN}\sum_i\mathbb{E}[\|c_i^t - \nabla f_i(x^t)\|^2] + \frac{4\alpha^2}{S}L^2 U^t + \frac{4\alpha^2\sigma^2}{SK}$$

$$\leq \frac{4\alpha^2}{SN}\sum_i\mathbb{E}[\|c_i^t - \nabla f_i(x^{t-1})\|^2] + \frac{4\alpha^2 L^2}{S}\mathbb{E}[\|x^t - x^{t-1}\|^2]$$

$$+ \frac{4\alpha^2}{S}L^2 U^t + \frac{4\alpha^2\sigma^2}{SK}. \tag{16}$$

Plugging (15) and (16) into (14), we obtain

$$\mathbb{E}[\|d^{t+1} - \nabla f(x^t)\|^2]$$

$$\leq 4\mathbb{E}[\|c^t - \nabla f(x^{t-1})\|^2] + \frac{4\alpha^2}{S}\frac{1}{N}\sum_i \mathbb{E}[\|c_i^t - \nabla f_i(x^{t-1})\|^2]$$

$$+ 4(1 + S^{-1})\alpha^2 L^2 U^t + 4\left(1 + \frac{\alpha^2}{S}\right)L^2\mathbb{E}[\|x^t - x^{t-1}\|^2] + (N^{-1} + S^{-1})\frac{4\alpha^2\sigma^2}{K}. \quad (17)$$

For $\|\tilde{d}^{t+1} - d^{t+1}\|^2$, using mutual independence and Definition 2, we have

$$\mathbb{E}[\|\tilde{d}^{t+1} - d^{t+1}\|^2] = \mathbb{E}\left[\left\|\frac{1}{S}\sum_{i\in\mathcal{S}^t}\mathcal{C}_i(\alpha(g_i^t - c_i^t)) - \alpha(g_i^t - c_i^t)\right\|^2\right]$$

$$\leq \frac{\omega\alpha^2}{S^2}\mathbb{E}\left[\sum_{i\in\mathcal{S}^t}\|g_i^t - c_i^t\|^2\right] = \frac{\omega\alpha^2}{S}\frac{1}{N}\sum_i \mathbb{E}[\|g_i^t - c_i^t\|^2].$$

Then applying the same relaxation in (16), we obtain

$$\mathbb{E}[\|\tilde{d}^{t+1} - d^{t+1}\|^2] \leq \frac{4\omega\alpha^2}{SN}\sum_i \mathbb{E}[\|c_i^t - \nabla f_i(x^{t-1})\|^2] + \frac{4\omega\alpha^2 L^2}{S}\mathbb{E}[\|x^t - x^{t-1}\|^2]$$

$$+ \frac{4\omega\alpha^2}{S}U^t + \frac{4\omega\alpha^2\sigma^2}{SK}. \quad (18)$$

Plugging (17) and (18) into (13) and noting $N^{-1} \leq S^{-1}$, we complete the proof. $\quad\square$

Given Lemma 4, the rest is to bound $\|c^t - \nabla f(x^{t-1})\|^2$, $\|c_i^t - \nabla f_i(x^{t-1})\|^2$.

**Lemma 5.** *Under Assumptions 1 and 2, it holds for all $t \geq 0$ that*

$$\mathbb{E}[\|c^{t+1} - \nabla f(x^t)\|^2]$$

$$\leq \left(1 - \frac{S\alpha}{2N}\right)\mathbb{E}[\|c^t - \nabla f(x^t)\|^2] + \frac{4(1+\omega)\alpha^2 S}{N^2}\frac{1}{N}\sum_i \mathbb{E}[\|c_i^t - \nabla f_i(x^{t-1})\|^2]$$

$$+ \left(\frac{2N}{S\alpha} + \frac{4(1+\omega)\alpha^2 S}{N^2}\right)L^2\mathbb{E}[\|x^t - x^{t-1}\|^2]$$

$$+ \left(\frac{2S}{N} + \frac{4(1+\omega)\alpha S}{N^2}\right)\alpha L^2 U^t + \frac{6(1+\omega)\alpha^2 S\sigma^2}{N^2 K}. \quad (19)$$

*Proof.* Using (8) and Lemma 1, we have

$$\mathbb{E}[\|c^{t+1} - \nabla f(x^t)\|^2]$$

$$= \mathbb{E}\left[\left\|\frac{1}{S}\sum_{i\in\mathcal{S}^t}\frac{S}{N}\mathcal{C}_i(\alpha(g_i^t - c_i^t)) + c^t - \nabla f(x^t)\right\|^2\right]$$

$$\leq \mathbb{E}\left[\left\|\frac{1}{S}\sum_{i\in\mathcal{S}^t}\frac{S\alpha}{N}(g_i^t - c_i^t) + c^t - \nabla f(x^t)\right\|^2\right] + \frac{\omega\alpha^2}{N^2}\sum_{i\in\mathcal{S}^t}\mathbb{E}[\|g_i^t - c_i^t\|^2]$$

$$\leq \mathbb{E}\left[\left\|\frac{S\alpha}{N}(g^t - c^t) + c^t - \nabla f(x^t)\right\|^2\right] + \frac{(1+\omega)\alpha^2 S}{N^2}\frac{1}{N}\sum_i \mathbb{E}[\|g_i^t - c_i^t\|^2] \quad (20)$$

Using (10), Young's inequality, and Assumption 1, we further have

$$\mathbb{E}\left[\left\|\frac{S\alpha}{N}(g^t - c^t) + c^t - \nabla f(x^t)\right\|^2\right]$$

$$\leq \left(1 - \frac{S\alpha}{N}\right)\mathbb{E}[\|c^t - \nabla f(x^t)\|^2] + \frac{2\alpha SL^2}{N}U^t + \frac{2\alpha^2 S^2\sigma^2}{N^3 K}$$

$$\leq \left(1 - \frac{S\alpha}{2N}\right)\mathbb{E}[\|c^t - \nabla f(x^{t-1})\|^2] + \frac{2NL^2}{S\alpha}\mathbb{E}[\|x^t - x^{t-1}\|^2] + \frac{2\alpha SL^2}{N}U^t + \frac{2\alpha^2 S^2\sigma^2}{N^3 K}. \tag{21}$$

Using Young's inequality and Assumption 1, we can obtain

$$\frac{(1+\omega)\alpha^2 S}{N^2}\frac{1}{N}\sum_i \mathbb{E}[\|g_i^t - c_i^t\|^2]$$

$$\leq \frac{(1+\omega)\alpha^2 S}{N^2}\frac{1}{N}\sum_i \left(2\mathbb{E}[\|c_i^t - \nabla f_i(x^t)\|^2] + 2\mathbb{E}[\|g_i^t - \nabla f_i(x^t)\|^2]\right)$$

$$\leq \frac{(1+\omega)\alpha^2 S}{N^2}\frac{1}{N}\sum_i \left(4\mathbb{E}[\|c_i^t - \nabla f_i(x^{t-1})\|^2] + 4L^2\mathbb{E}[\|x^t - x^{t-1}\|^2] + 2\mathbb{E}[\|g_i^t - \nabla f_i(x^t)\|^2]\right). \tag{22}$$

Using (11), we have

$$\frac{1}{N}\sum_i \mathbb{E}[\|g_i^t - \nabla f_i(x^t)\|^2] \leq 2L^2 U^t + \frac{2\sigma^2}{K}. \tag{23}$$

Plugging (23) into (22), we reach

$$\frac{(1+\omega)\alpha^2 S}{N^2}\frac{1}{N}\sum_i \mathbb{E}[\|g_i^t - c_i^t\|^2]$$

$$\leq \frac{(1+\omega)\alpha^2 S}{N^2}\frac{1}{N}\sum_i \left(4\mathbb{E}[\|c_i^t - \nabla f_i(x^{t-1})\|^2] + 4L^2\mathbb{E}[\|x^t - x^{t-1}\|^2] + 4L^2 U^t + \frac{4\sigma^2}{K}\right) \tag{24}$$

Combining (20), (21), (24) together and using $\frac{\alpha^2 S^2\sigma^2}{N^3 K} \leq \frac{(1+\omega)\alpha^2 S\sigma^2}{N^2 K}$ completes the proof. $\qquad\square$

**Lemma 6.** *Under Assumptions 1 and 2, suppose $0 \leq \alpha \leq \frac{1}{4(\omega+1)}$, then it holds for all $t \geq 0$ that*

$$\frac{1}{N}\sum_i \mathbb{E}[\|c_i^{t+1} - \nabla f_i(x^t)\|^2]$$

$$\leq \left(1 - \frac{S\alpha}{4N}\right)\frac{1}{N}\sum_i \mathbb{E}[\|c_i^t - \nabla f_i(x^{t-1})\|^2] + \frac{4NL^2}{S\alpha}\mathbb{E}[\|x^t - x^{t-1}\|^2]$$

$$+ \frac{3\alpha L^2 S}{N}U^t + \frac{2(1+\omega)\alpha^2 S\sigma^2}{NK}. \tag{25}$$

*Proof.* Using (8), we have

$$\frac{1}{N}\sum_i \mathbb{E}[\|c_i^{t+1} - \nabla f_i(x^t)\|^2]$$

$$= \frac{1}{N}\sum_i \left(\left(1 - \frac{S}{N}\right)\mathbb{E}[\|c_i^t - \nabla f_i(x^t)\|^2] + \frac{S}{N}\mathbb{E}[\|\mathcal{C}_i(\alpha(g_i^t - c_i^t)) + c_i^t - \nabla f_i(x^t)\|^2]\right)$$

$$\leq \frac{1}{N}\sum_i \left(\left(1 - \frac{S}{N}\right)\mathbb{E}[\|c_i^t - \nabla f_i(x^t)\|^2] + \frac{S}{N}\mathbb{E}[\|\alpha(g_i^t - c_i^t) + c_i^t - \nabla f_i(x^t)\|^2]\right.$$

$$\left. + \frac{S\omega\alpha^2}{N}\mathbb{E}[\|g_i^t - c_i^t\|^2]\right) \tag{26}$$

Note that

$$\frac{S}{N}\frac{1}{N}\sum_i \mathbb{E}[\|\alpha(g_i^t - c_i^t) + c_i^t - \nabla f_i(x^t)\|^2]$$

$$=\frac{S}{N}\frac{1}{N}\sum_i \mathbb{E}[\|\alpha(g_i^t - \nabla f_i(x^t)) + (1-\alpha)(c_i^t - \nabla f_i(x^t))\|^2]$$

$$\leq\frac{S}{N}\left(\frac{1-\alpha}{N}\sum_i \mathbb{E}[\|c_i^t - \nabla f_i(x^t)\|^2] + 2\alpha L^2 U^t + \frac{2\alpha^2\sigma^2}{K}\right) \tag{27}$$

and by applying (11),

$$\frac{\omega\alpha^2 S}{N}\frac{1}{N}\sum_i \mathbb{E}[\|g_i^t - c_i^t\|^2] = \frac{4\omega\alpha^2 S}{N}\frac{1}{N}\sum_i \mathbb{E}\left[\left\|\frac{1}{2}(g_i^t - \nabla f_i(x^t)) - \frac{1}{2}(\nabla f_i(x^t) - c_i^t)\right\|^2\right]$$

$$\leq\frac{4\omega\alpha^2 S}{N}\left(\frac{1}{2N}\sum_i \mathbb{E}[\|\nabla f_i(x^t) - c_i^t\|^2] + L^2 U^t + \frac{\sigma^2}{2K}\right). \tag{28}$$

Plugging (27) and (28) into (26), we obtain

$$\frac{1}{N}\sum_i \mathbb{E}[\|c_i^{t+1} - \nabla f_i(x^t)\|^2]$$

$$\leq\left(1 - \frac{S\alpha(1-2\omega\alpha)}{N}\right)\frac{1}{N}\sum_i \mathbb{E}[\|c_i^t - \nabla f_i(x^t)\|^2] + \frac{(2+4\omega\alpha)\alpha SL^2}{N}U^t + \frac{2(1+\omega)\alpha^2 S\sigma^2}{NK}$$

$$\leq\left(1 - \frac{S\alpha}{2N}\right)\frac{1}{N}\sum_i \mathbb{E}[\|c_i^t - \nabla f_i(x^t)\|^2] + \frac{3\alpha SL^2}{N}U^t + \frac{2(1+\omega)\alpha^2 S\sigma^2}{NK}$$

where we use $\alpha \leq \frac{1}{4(\omega+1)}$ in the last inequality. By further using Young's inequality and Assumption 1, we obtain

$$\frac{1}{N}\sum_i \mathbb{E}[\|c_i^{t+1} - \nabla f_i(x^t)\|^2]$$

$$\leq\left(1 - \frac{S\alpha}{4N}\right)\frac{1}{N}\sum_i \mathbb{E}[\|c_i^t - \nabla f_i(x^{t-1})\|^2] + \frac{4NL^2}{S\alpha}\mathbb{E}[\|x^t - x^{t-1}\|^2]$$

$$+ \frac{3\alpha L^2 S}{N}U^t + \frac{2(1+\omega)\alpha^2 S\sigma^2}{NK}.$$

$$\square$$

**Lemma 7.** *Under Assumptions 1 and 2, it holds for any $t \geq 0$ and $\eta_l KL \leq \frac{1}{2}$ that*

$$U^t \leq \frac{9e^2 K^2\eta_l^2}{N}\sum_i \left(\mathbb{E}[\|c_i^t - \nabla f_i(x^{t-1})\|^2] + L^2\mathbb{E}[\|x^t - x^{t-1}\|^2] + \mathbb{E}[\|\nabla f(x^t)\|^2]\right) + e^2 K\eta_l^2\sigma^2.$$

*Proof.* The proof is similar to that of Lemma 12. $\square$

**Theorem 3.** *Under Assumptions 1 and 2, suppose clients are associated with mutually independent $\omega$-unbiased compressors, if we initialize $c_i^0 = \frac{1}{B}\sum_{b=1}^B \nabla F(x^0; \xi_i^b)$, $c^0 = \frac{1}{N}\sum_{i=1}^N c_i^0$ with $\{\xi_i^b\}_{b=1}^B \overset{iid}{\sim} \mathcal{D}_i$ and $B \gtrsim \frac{\sigma^2}{NL\Delta}$ ($c_i^0 \to \nabla f_i(x^0)$ as $B \to \infty$), set*

$$\eta_l KL \leq \sqrt{\frac{\alpha(1+\omega)}{1400e^2 N}}, \quad \eta_g\eta_l KL = \frac{27\alpha S}{N},$$

$$\alpha = \left(4(1+\omega) + \left(\frac{(1+\omega)TS\sigma^2}{N^2 KL\Delta}\right)^{1/2} + \left(\frac{(1+\omega)T\sigma^2}{NKL\Delta}\right)^{1/3}\right)^{-1}, \tag{29}$$

*then* SCALLION *converges as*

$$\frac{1}{T} \sum_{t=0}^{T-1} \mathbb{E}[\|\nabla f(x^t)\|^2] \lesssim \sqrt{\frac{(1+\omega)L\Delta\sigma^2}{SKT}} + \left(\frac{(1+\omega)N^2 L^2 \Delta^2 \sigma^2}{S^3 KT^2}\right)^{1/3} + \frac{(1+\omega)NL\Delta}{ST}$$

*where* $\Delta \triangleq f(x^0) - \min_x f(x)$.

*Proof.* Adding (19) $\times \frac{10\gamma N}{\alpha S}$ to (12), we have

$$\mathbb{E}[f(x^{t+1})] + \frac{10\gamma N}{\alpha S} \mathbb{E}[\|c^{t+1} - \nabla f(x^t)\|^2]$$

$$\leq \mathbb{E}[f(x^t)] + \gamma \left(\frac{10N}{\alpha S} - 1\right) \mathbb{E}[\|c^t - \nabla f(x^{t-1})\|^2] - \frac{\gamma}{2} \mathbb{E}[\|\nabla f(x^t)\|^2]$$

$$- \left(\frac{1}{2\gamma} - \frac{L}{2}\right) \mathbb{E}[\|x^{t+1} - x^t\|^2] + \gamma L^2 \left(\frac{44(1+\omega)\alpha}{S} + \frac{24N^2}{\alpha^2 S^2}\right) \mathbb{E}[\|x^t - x^{t-1}\|^2]$$

$$+ \gamma(1+\omega) \left(\frac{8\alpha^2}{S} + \frac{60\alpha}{N}\right) \frac{\sigma^2}{K} + \gamma L^2 \left(24 + \frac{4(1+\omega)\alpha^2}{S} + \frac{40(1+\omega)\alpha}{N}\right) U^t$$

$$+ \frac{40\gamma(1+\omega)\alpha}{N} \frac{1}{N} \sum_i \mathbb{E}[\|c_i^t - \nabla f_i(x^{t-1})\|^2]. \tag{30}$$

Adding (25) $\times \frac{164\gamma(1+\omega)}{S}$ to (30), we have

$$\mathbb{E}[f(x^{t+1})] + \frac{10\gamma N}{\alpha S} \mathbb{E}[\|c^{t+1} - \nabla f(x^t)\|^2] + \frac{164\gamma(1+\omega)}{S} \frac{1}{N} \sum_i \mathbb{E}[\|c_i^{t+1} - \nabla f_i(x^t)\|^2]$$

$$\leq \mathbb{E}[f(x^t)] + \gamma \left(\frac{10N}{\alpha S} - 1\right) \mathbb{E}[\|c^t - \nabla f(x^{t-1})\|^2]$$

$$+ \left(\frac{164\gamma(1+\omega)}{S} - \frac{\gamma(1+\omega)}{N}\right) \frac{1}{N} \sum_i \mathbb{E}[\|c_i^t - \nabla f_i(x^{t-1})\|^2]$$

$$- \frac{\gamma}{2} \mathbb{E}[\|\nabla f(x^t)\|^2] - \left(\frac{1}{2\gamma} - \frac{L}{2}\right) \mathbb{E}[\|x^{t+1} - x^t\|^2]$$

$$+ \gamma L^2 \left(\frac{44(1+\omega)\alpha}{S} + \frac{24N^2}{\alpha^2 S^2} + \frac{656(1+\omega)N}{\alpha S^2}\right) \mathbb{E}[\|x^t - x^{t-1}\|^2]$$

$$+ \gamma(1+\omega) \left(\frac{8\alpha^2}{S} + \frac{60\alpha}{N} + \frac{328(1+\omega)\alpha^2}{N}\right) \frac{\sigma^2}{K}$$

$$+ \gamma L^2 \left(24 + \frac{4(1+\omega)\alpha^2}{S} + \frac{522(1+\omega)\alpha}{N}\right) U^t. \tag{31}$$

Defining the Lyapunov function ($x^{-1} := x^0$)

$$\Phi^t := \mathbb{E}[f(x^t)] + \frac{10\gamma N}{\alpha S} \mathbb{E}[\|c^t - \nabla f(x^{t-1})\|^2] + \frac{164\gamma(1+\omega)}{S} \frac{1}{N} \sum_i \mathbb{E}[\|c_i^t - \nabla f_i(x^{t-1})\|^2],$$

following (31), we obtain

$$\Phi^{t+1} - \Phi^t$$

$$\leq -\frac{\gamma}{2}\mathbb{E}[\|\nabla f(x^t)\|^2] - \gamma\mathbb{E}[\|c^t - \nabla f(x^{t-1})\|^2] - \frac{\gamma(1+\omega)}{N}\frac{1}{N}\sum_i \mathbb{E}[\|c_i^t - \nabla f_i(x^{t-1})\|^2]$$

$$- \left(\frac{1}{2\gamma} - \frac{L}{2}\right)\mathbb{E}[\|x^{t+1} - x^t\|^2]$$

$$+ \gamma L^2 \left(\frac{44(1+\omega)\alpha}{S} + \frac{24N^2}{\alpha^2 S^2} + \frac{656(1+\omega)N}{\alpha S^2}\right)\mathbb{E}[\|x^t - x^{t-1}\|^2]$$

$$+ \gamma(1+\omega)\left(\frac{8\alpha^2}{S} + \frac{60\alpha}{N} + \frac{328(1+\omega)\alpha^2}{N}\right)\frac{\sigma^2}{K}$$

$$+ \gamma L^2 \left(24 + \frac{4(1+\omega)\alpha^2}{S} + \frac{522(1+\omega)\alpha}{N}\right)U^t. \tag{32}$$

Since $N \geq S \geq 1$ and $\alpha \leq 1/(4(1+\omega))$, we have

$$9e^2 K^2 \eta_l^2 L^2 \left(24 + \frac{4(1+\omega)\alpha^2}{S} + \frac{522(1+\omega)\alpha}{N}\right)$$

$$\leq 9e^2 K^2 \eta_l^2 L^2 \left(24 + \frac{1}{4} + \frac{261}{2}\right) \leq 1400e^2 K^2 \eta_l^2 L^2 \leq \frac{\alpha(1+\omega)}{N} \leq \frac{1}{4}. \tag{33}$$

Combining (33) with Lemma 7, we have

$$\gamma L^2 \left(24 + \frac{4(1+\omega)\alpha^2}{S} + \frac{522(1+\omega)\alpha}{N}\right)U^t$$

$$\leq \gamma\mathbb{E}[\|c^t - \nabla f(x^{t-1})\|^2] + \frac{\gamma(1+\omega)}{N}\frac{1}{N}\sum_i \mathbb{E}[\|c_i^t - \nabla f_i(x^{t-1})\|^2]$$

$$+ \frac{\gamma}{4}\mathbb{E}[\|\nabla f(x^t)\|^2] + \frac{\gamma L^2}{4}\mathbb{E}[\|x^t - x^{t-1}\|^2] + \frac{\gamma\alpha(1+\omega)\sigma^2}{NK}. \tag{34}$$

Plugging (34) into (32), we reach

$$\Phi^{t+1} - \Phi^t$$

$$\leq -\frac{\gamma}{4}\mathbb{E}[\|\nabla f(x^t)\|^2] + \gamma(1+\omega)\left(\frac{8\alpha^2}{S} + \frac{60\alpha}{N} + \frac{328(1+\omega)\alpha^2}{N}\right)\frac{\sigma^2}{K}$$

$$- \left(\frac{1}{2\gamma} - \frac{L}{2}\right)\mathbb{E}[\|x^{t+1} - x^t\|^2]$$

$$+ \gamma L^2 \left(\frac{44(1+\omega)\alpha}{S} + \frac{25N^2}{\alpha^2 S^2} + \frac{656(1+\omega)N}{\alpha S^2}\right)\mathbb{E}[\|x^t - x^{t-1}\|^2]. \tag{35}$$

Due to the choice of $\gamma = \eta_g \eta_l K$ and $\alpha \leq \frac{1}{4(\omega+1)}$, it holds that

$$\frac{1}{2\gamma} \geq \frac{L}{2} + \gamma L^2 \left(\frac{44(1+\omega)\alpha}{S} + \frac{25N^2}{\alpha^2 S^2} + \frac{656(1+\omega)N}{\alpha S^2}\right).$$

Averaging (35) over $k$ and noting $\|x^0 - x^{-1}\|^2 = 0$, $\alpha = O((1+\omega)^{-1})$, we obtain

$$\frac{1}{T}\sum_{t=0}^{T-1}\mathbb{E}[\|\nabla f(x^t)\|^2] \lesssim \frac{\Phi^0 - \Phi^T}{\gamma T} + (1+\omega)\left(\frac{\alpha^2}{S} + \frac{\alpha}{N} + \frac{(1+\omega)\alpha^2}{N}\right)\frac{\sigma^2}{K}$$

$$\lesssim \frac{\Phi^0 - \Phi^T}{\gamma T} + (1+\omega)\left(\frac{\alpha^2}{S} + \frac{\alpha}{N}\right)\frac{\sigma^2}{K}.$$

By the definition of $\Phi^t$, it holds that

$$\frac{\Phi^0 - \Phi^T}{\gamma T} \lesssim \frac{L\Delta}{\gamma T} + \frac{N}{\alpha S} \frac{\mathbb{E}[\|c^0 - \nabla f(x^0)\|^2]}{T} + \frac{(1+\omega)\frac{1}{N}\sum_i \mathbb{E}[\|c_i^0 - \nabla f_i(x^0)\|^2]}{S} \frac{}{T}$$

$$\lesssim \frac{L\Delta}{T}\frac{N}{\alpha S} + \frac{\sigma^2}{\alpha SBT}$$

where we use the choice of $\gamma$, $\alpha$, and the initialization of $\{c_i^0\}_{i\in[N]}$ and $c^0$ in the second inequality. Due to the choice of $B$, we have $\frac{\sigma^2}{\alpha SBT} \lesssim \frac{L\Delta}{T}\frac{N}{\alpha S}$ and thus

$$\frac{1}{T}\sum_{t=0}^{T-1}\mathbb{E}[\|\nabla f(x^t)\|^2] \lesssim \frac{L\Delta}{T}\frac{N}{\alpha S} + (1+\omega)\left(\frac{\alpha^2}{S} + \frac{\alpha}{N}\right)\frac{\sigma^2}{K}. \tag{36}$$

Plugging the choice of $\alpha$ into (36) completes the proof. $\qquad\square$

# F  PROOF OF SCAFCOM

In this section, we prove the convergence result of SCAFCOM with biased compression, where we additionally define $x^{-1} := x^0$. We thus have $\mathbb{E}[\|x^t - x^{t-1}\|^2] = 0$ for $t = 0$. Note that $x^{-1}$ is defined for the purpose of notation and is not utilized in our algorithms.

**Lemma 8** (DESCENT LEMMA). *Under Assumptions 1 and 2, it holds for all $t \geq 0$ and $\gamma > 0$ that*

$$\mathbb{E}[f(x^{t+1})]$$

$$\leq \mathbb{E}[f(x^t)] - \frac{\gamma}{2}\mathbb{E}[\|\nabla f(x^t)\|^2] - \left(\frac{1}{2\gamma} - \frac{L}{2}\right)\mathbb{E}[\|x^{t+1} - x^t\|^2] + 16\gamma L^2 \mathbb{E}[\|x^t - x^{t-1}\|^2]$$

$$+ \gamma\left(4\mathbb{E}[\|v^t - \nabla f(x^{t-1})\|^2] + \frac{12\beta^2}{N}\sum_i \mathbb{E}[\|v_i^t - \nabla f_i(x^{t-1})\|^2] + \frac{12}{N}\sum_i \mathbb{E}[\|v_i^t - c_i^t\|^2]\right)$$

$$+ \frac{8\gamma\beta^2\sigma^2}{K} + 9\gamma\beta^2 L^2 U^t. \tag{37}$$

*Proof.* By Lemma 2 of Li et al. (2021), we have

$$f(x^{t+1}) \leq f(x^t) - \frac{\gamma}{2}\|\nabla f(x^t)\|^2 - \left(\frac{1}{2\gamma} - \frac{L}{2}\right)\|x^{t+1} - x^t\|^2 + \frac{\gamma}{2}\|\tilde{d}^{t+1} - \nabla f(x^t)\|^2$$

where $\tilde{d}^{t+1} = \frac{1}{S}\sum_{i\in\mathcal{S}^t}\tilde{\delta}_i^t + c^t$. Letting $d^{t+1} \triangleq \frac{1}{S}\sum_{i\in\mathcal{S}^t}\delta_i^t + c^t$, we further have

$$f(x^{t+1}) \leq f(x^t) - \frac{\gamma}{2}\|\nabla f(x^t)\|^2 - \left(\frac{1}{2\gamma} - \frac{L}{2}\right)\|x^{t+1} - x^t\|^2$$

$$+ \gamma\|\tilde{d}^{t+1} - d^{t+1}\|^2 + \gamma\|d^{t+1} - \nabla f(x^t)\|^2. \tag{38}$$

For $\|d^{t+1} - \nabla f(x^t)\|^2$, using Lemma 2 and the fact that $c^t \equiv \frac{1}{N}\sum_i c_i^t$ and $d^{t+1} = \frac{1}{S}\sum_{i\in\mathcal{S}^t}(u_i^{t+1} - c_i^t) + c^t$, we have

$$\mathbb{E}[\|d^{t+1} - \nabla f(x^t)\|^2]$$

$$= \mathbb{E}\left[\left\|c^t + \frac{1}{S}\sum_{i\in\mathcal{S}^t}(u_i^{t+1} - c_i^t) - \nabla f(x^t)\right\|^2\right]$$

$$\leq \mathbb{E}[\|u^{t+1} - \nabla f(x^t)\|^2] + \frac{1}{SN}\sum_i \mathbb{E}[\|u_i^{t+1} - c_i^t\|^2]$$

$$= \mathbb{E}[\|(1-\beta)(v^t - \nabla f(x^t)) + \beta(g^t - \nabla f(x^t))\|^2] + \frac{1}{SN}\sum_i \mathbb{E}[\|v_i^t + \beta(g_i^t - v_i^t) - c_i^t\|^2]. \tag{39}$$

Using (10) and Assumption 1, we have

$$\mathbb{E}[\|(1-\beta)(v^t - \nabla f(x^t)) + \beta(g^t - \nabla f(x^t)\|^2]$$

$$\leq 2\mathbb{E}[\|v^t - \nabla f(x^t)\|^2] + 3\beta^2 L^2 U^t + \frac{2\beta^2\sigma^2}{NK}$$

$$\leq 4\mathbb{E}[\|v^t - \nabla f(x^{t-1})\|^2] + 4L^2\mathbb{E}[\|x^t - x^{t-1}\|^2] + 3\beta^2 L^2 U^t + \frac{2\beta^2\sigma^2}{NK}. \tag{40}$$

Similarly, using (11) and Assumption 1, we have

$$\frac{1}{SN} \sum_i \mathbb{E}[\|v_i^t + \beta(g_i^t - v_i^t) - c_i^t\|^2]$$

$$= \frac{1}{SN} \sum_i \mathbb{E}[\|v_i^t + \beta(\nabla f_i(x^t) - \nabla f_i(x^{t-1})) + \beta(\nabla f_i(x^{t-1}) - v_i^t) + \beta(g_i^t - \nabla f_i(x^t)) - c_i^t\|^2]$$

$$\leq \frac{2}{SN} \sum_i \mathbb{E}[\|v_i^t + \beta(\nabla f_i(x^t) - \nabla f_i(x^{t-1})) + \beta(\nabla f_i(x^{t-1}) - v_i^t) - c_i^t\|^2] + \frac{3\beta^2 L^2 U^t}{S} + \frac{2\beta^2 \sigma^2}{SK}$$

$$\leq \frac{6}{SN} \sum_i \left( \mathbb{E}[\|v_i^t - c_i^t\|^2] + \beta^2 \mathbb{E}[\|v_i^t - \nabla f_i(x^{t-1})\|^2] + \beta^2 L^2 \mathbb{E}[\|x^t - x^{t-1}\|^2] \right) + \frac{3\beta^2 L^2 U^t}{S} + \frac{2\beta^2 \sigma^2}{SK}. \tag{41}$$

Plugging (40) and (41) into (39), we obtain

$$\mathbb{E}[\|d^{t+1} - \nabla f(x^t)\|^2]$$

$$\leq 4\mathbb{E}[\|v^t - \nabla f(x^{t-1})\|^2] + \frac{6\beta^2}{S} \frac{1}{N} \sum_i \mathbb{E}[\|v_i^t - \nabla f_i(x^{t-1})\|^2] + \frac{6}{SN} \sum_i \mathbb{E}[\|v_i^t - c_i^t\|^2]$$

$$+ 3(1 + S^{-1})\beta^2 L^2 U^t + \left( 4 + \frac{6\beta^2}{S} \right) L^2 \mathbb{E}[\|x^t - x^{t-1}\|^2] + (N^{-1} + S^{-1})\frac{2\beta^2 \sigma^2}{K}. \tag{42}$$

For $\|\tilde{d}^{t+1} - d^{t+1}\|^2$, using Young's inequality and Definition 2, we have

$$\mathbb{E}[\|\tilde{d}^{t+1} - d^{t+1}\|^2] = \mathbb{E}\left[ \left\| \frac{1}{S} \sum_{i \in \mathcal{S}^t} \mathcal{C}_i(u_i^{t+1} - c_i^t) - (u_i^{t+1} - c_i^t) \right\|^2 \right]$$

$$\leq \frac{q^2}{S} \mathbb{E}\left[ \sum_{i \in \mathcal{S}^t} \|u_i^{t+1} - c_i^t\|^2 \right] = \frac{q^2}{N} \sum_i \mathbb{E}[\|u_i^{t+1} - c_i^t\|^2]$$

$$= \frac{q^2}{N} \sum_i \mathbb{E}[\|v_i^t + \beta(g_i^t - v_i^t) - c_i^t\|^2].$$

Then applying the same relaxation in (41), we obtain

$$\mathbb{E}[\|\tilde{d}^{t+1} - d^{t+1}\|^2]$$

$$\leq \frac{6q^2}{N} \sum_i \left( \mathbb{E}[\|v_i^t - c_i^t\|^2] + \beta^2 \mathbb{E}[\|v_i^t - \nabla f_i(x^{t-1})\|^2] + \beta^2 L^2 \mathbb{E}[\|x^t - x^{t-1}\|^2] \right)$$

$$+ 3\beta^2 q^2 L^2 U^t + \frac{2\beta^2 q^2 \sigma^2}{K}. \tag{43}$$

Plugging (42) and (43) into (38) and noting $N^{-1} \leq S^{-1} \leq 1$, $q^2 \leq 1$, we complete the proof. $\qquad\square$

Given Lemma 8, the rest is to bound $\|v^t - \nabla f(x^{t-1})\|^2$, $\|v_i^t - \nabla f_i(x^{t-1})\|^2$, and $\|v_i^t - c_i^t\|^2$.

**Lemma 9.** *Under Assumptions 1 and 2, it holds for all $t \geq 0$ that*

$$\mathbb{E}[\|v^{t+1} - \nabla f(x^t)\|^2]$$

$$\leq \left( 1 - \frac{S\beta}{2N} \right) \mathbb{E}[\|v^t - \nabla f(x^{t-1})\|^2] + \frac{4\beta^2 S}{N^2} \frac{1}{N} \sum_i \mathbb{E}[\|v_i^t - \nabla f_i(x^{t-1})\|^2]$$

$$+ \frac{6NL^2}{S\beta} \mathbb{E}[\|x^t - x^{t-1}\|^2] + \frac{6\beta SL^2}{N} U^t + \frac{6\beta^2 S\sigma^2}{N^2 K}. \tag{44}$$

*Proof.* Using (9) and Lemma 1, we have

$$\mathbb{E}[\|v^{t+1} - \nabla f(x^t)\|^2]$$

$$= \mathbb{E}\left[\left\|\frac{1}{S}\sum_{i\in\mathcal{S}^t}\frac{S\beta}{N}(g_i^t - v_i^t) + v^t - \nabla f(x^t)\right\|^2\right]$$

$$\leq \mathbb{E}\left[\left\|\frac{S\beta}{N}(g^t - v^t) + v^t - \nabla f(x^t)\right\|^2\right] + \frac{\beta^2 S}{N^2}\frac{1}{N}\sum_i \mathbb{E}[\|g_i^t - v_i^t\|^2]$$

$$= \mathbb{E}\left[\left\|\left(1 - \frac{S\beta}{N}\right)(v^t - \nabla f(x^t)) + \frac{S\beta}{N}(g^t - \nabla f(x^t))\right\|^2\right] + \frac{\beta^2 S}{N^2}\frac{1}{N}\sum_i \mathbb{E}[\|g_i^t - v_i^t\|^2]. \quad (45)$$

Using (10), Young's inequality, and Assumption 1, we further have

$$\mathbb{E}\left[\left\|\left(1 - \frac{S\beta}{N}\right)(v^t - \nabla f(x^t)) + \frac{S\beta}{N}(g^t - \nabla f(x^t))\right\|^2\right]$$

$$\leq \left(1 - \frac{S\beta}{N}\right)\mathbb{E}[\|v^t - \nabla f(x^t)\|^2] + \frac{2\beta S L^2}{N}U^t + \frac{2\beta^2 S^2 \sigma^2}{N^3 K}$$

$$\leq \left(1 - \frac{S\beta}{2N}\right)\mathbb{E}[\|v^t - \nabla f(x^{t-1})\|^2] + \frac{2N L^2}{S\beta}\mathbb{E}[\|x^t - x^{t-1}\|^2] + \frac{2\beta S L^2}{N}U^t + \frac{2\beta^2 S^2 \sigma^2}{N^3 K}. \quad (46)$$

Using Young's inequality and Assumption 1, we can obtain

$$\frac{\beta^2 S}{N^2}\frac{1}{N}\sum_i \mathbb{E}[\|g_i^t - v_i^t\|^2]$$

$$\leq \frac{\beta^2 S}{N^2}\frac{1}{N}\sum_i \left(2\mathbb{E}[\|\nabla f_i(x^t) - v_i^t\|^2] + 2\mathbb{E}[\|g_i^t - \nabla f_i(x^t)\|^2]\right)$$

$$\leq \frac{\beta^2 S}{N^2}\frac{1}{N}\sum_i \left(4\mathbb{E}[\|v_i^t - \nabla f_i(x^{t-1})\|^2] + 4L^2\mathbb{E}[\|x^t - x^{t-1}\|^2] + 2\mathbb{E}[\|g_i^t - \nabla f_i(x^t)\|^2]\right). \quad (47)$$

Using Lemma 2 and Assumption 1, we have

$$\frac{1}{N}\sum_i \mathbb{E}[\|g_i^t - \nabla f_i(x^t)\|^2]$$

$$\leq \frac{2}{N}\sum_i \mathbb{E}\left[\left\|\frac{1}{K}\sum_k \nabla f_i(y_i^{t,k}) - \nabla f_i(x^t)\right\|^2\right] + \frac{2\sigma^2}{K} \leq 2L^2 U^t + \frac{2\sigma^2}{K}. \quad (48)$$

Plugging (48) into (47), we reach

$$\frac{\beta^2 S}{N^2}\frac{1}{N}\sum_i \mathbb{E}[\|g_i^t - v_i^t\|^2]$$

$$\leq \frac{\beta^2 S}{N^2}\frac{1}{N}\sum_i \left(4\mathbb{E}[\|v_i^t - \nabla f_i(x^{t-1})\|^2] + 4L^2\mathbb{E}[\|x^t - x^{t-1}\|^2] + 4L^2 U^t + \frac{4\sigma^2}{K}\right). \quad (49)$$

Combining (45), (46), (49) together and using $\frac{\beta^2 S^2 \sigma^2}{N^3 K} \leq \frac{\beta^2 S \sigma^2}{N^2 K}$, $\frac{\beta^2 S L^2}{N^2} \leq \frac{N L^2}{S\beta}$, $\frac{\beta^2 S L^2}{N^2} \leq \frac{\beta S L^2}{N}$ completes the proof. $\qquad\square$

**Lemma 10.** *Under Assumptions 1 and 2, it holds for all $t \geq 0$ that*

$$\frac{1}{N}\sum_i \mathbb{E}[\|v_i^{t+1} - \nabla f_i(x^t)\|^2]$$

$$\leq \left(1 - \frac{S\beta}{2N}\right)\frac{1}{N}\sum_i \mathbb{E}[\|v_i^t - \nabla f_i(x^{t-1})\|^2] + \frac{2N L^2}{S\beta}\mathbb{E}[\|x^t - x^{t-1}\|^2]$$

$$+ \frac{2\beta L^2 S}{N}U^t + \frac{2\beta^2 S \sigma^2}{NK}. \quad (50)$$

*Proof.* Using (9), we have

$$\frac{1}{N}\sum_i \mathbb{E}[\|v_i^{t+1} - \nabla f_i(x^t)\|^2]$$

$$=\frac{1}{N}\sum_i\left(\left(1-\frac{S}{N}\right)\mathbb{E}[\|v_i^t - \nabla f_i(x^t)\|^2] + \frac{S}{N}\mathbb{E}\left[\left\|\beta(g_i^t - v_i^t) + v_i^t - \nabla f_i(x^t)\right\|^2\right]\right)$$

$$\leq\left(1-\frac{S\beta}{N}\right)\frac{1}{N}\sum_i\mathbb{E}[\|v_i^t - \nabla f_i(x^t)\|^2] + \frac{2\beta L^2 S}{N}U^t + \frac{2\beta^2 S\sigma^2}{NK}$$

where the inequality due to

$$\frac{1}{N}\sum_i\mathbb{E}\left[\left\|\beta(g_i^t - v_i^t) + v_i^t - \nabla f_i(x^t)\right\|^2\right]$$

$$=\frac{1}{N}\sum_i\mathbb{E}\left[\left\|\beta(g_i^t - \nabla f_i(x^t)) + (1-\beta)(v_i^t - \nabla f_i(x^t))\right\|^2\right]$$

$$\leq\frac{1-\beta}{N}\sum_i\mathbb{E}[\|v_i^t - \nabla f_i(x^t)\|^2] + 2\beta L^2 U^t + \frac{2\beta^2\sigma^2}{K}$$

by applying (11). By further using Young's inequality and Assumption 1, we obtain

$$\frac{1}{N}\sum_i\mathbb{E}[\|v_i^{t+1} - \nabla f_i(x^t)\|^2]$$

$$\leq\left(1-\frac{S\beta}{2N}\right)\frac{1}{N}\sum_i\mathbb{E}[\|v_i^t - \nabla f_i(x^{t-1})\|^2] + \frac{2NL^2}{S\beta}\mathbb{E}[\|x^t - x^{t-1}\|^2]$$

$$+\frac{2\beta L^2 S}{N}U^t + \frac{2\beta^2 S\sigma^2}{NK}.$$

$\square$

**Lemma 11.** *Under Assumptions 1 and 2, it holds for all $t \geq 0$ that*

$$\frac{1}{N}\sum_i\mathbb{E}[\|v_i^{t+1} - c_i^{t+1}\|^2]$$

$$\leq\left(1-\frac{S(1-q)}{N}\right)\frac{1}{N}\sum_i\mathbb{E}[\|v_i^t - c_i^t\|^2] + \frac{4\beta^2 q^2 S}{(1-q)N}\frac{1}{N}\sum_i\mathbb{E}[\|v_i^t - \nabla f_i(x^{t-1})\|^2]$$

$$+\frac{4\beta^2 L^2 q^2 S}{(1-q)N}\mathbb{E}[\|x^t - x^{t-1}\|^2] + \frac{3\beta^2 q^2 SL^2}{(1-q)N}U^t + \frac{2\beta^2 q^2 S\sigma^2}{NK}. \tag{51}$$

*Proof.* Using (9) and Definition 2, we have

$$\mathbb{E}[\|v_i^{t+1} - c_i^{t+1}\|^2] = \left(1-\frac{S}{N}\right)\mathbb{E}[\|v_i^t - c_i^t\|^2] + \frac{S}{N}\mathbb{E}[\|u_i^{t+1} - \mathcal{C}_i(u_i^{t+1} - c_i^t) - c_i^t)\|^2]$$

$$\leq\left(1-\frac{S}{N}\right)\mathbb{E}[\|v_i^t - c_i^t\|^2] + \frac{S}{N}\frac{q^2}{N}\sum_i\mathbb{E}[\|u_i^{t+1} - c_i^t\|^2]$$

$$=\left(1-\frac{S}{N}\right)\mathbb{E}[\|v_i^t - c_i^t\|^2] + \frac{S}{N}\frac{q^2}{N}\sum_i\mathbb{E}[\|v_i^t + \beta(g_i^t - v_i^t) - c_i^t\|^2] \tag{52}$$

where $u_i^{t+1} \triangleq v_i^t + \beta(g_i^t - v_i^t)$. Using Lemma 2 and Assumption 1, we have

$$\frac{q^2}{N} \sum_i \mathbb{E}[\|v_i^t + \beta(g_i^t - v_i^t) - c_i^t\|^2] = \frac{q^2}{N} \sum_i \mathbb{E}[\|v_i^t + \beta(\nabla f_i(x^t) - v_i^t) + \beta(g_i^t - \nabla f_i(x^t)) - c_i^t\|^2]$$

$$= \frac{q^2}{N} \sum_i \left( \mathbb{E}[\|v_i^t - c_i^t + \beta(\nabla f_i(x^t) - v_i^t)\|^2] \right.$$

$$+ 2\beta \mathbb{E}\left[ \left\langle v_i^t - c_i^t + \beta(\nabla f_i(x^t) - v_i^t), \frac{1}{K}\sum_k \nabla f_i(y_i^{t,k}) - \nabla f_i(x^t) \right\rangle \right]$$

$$\left. + \beta^2 \mathbb{E}\left[ \left\| \frac{1}{K}\sum_k \nabla F(y_i^{t,k}; \xi_i^{t,k}) - \nabla f_i(x^t) \right\|^2 \right] \right)$$

$$\leq \frac{q^2}{N} \sum_i \left( \mathbb{E}[\|v_i^t - c_i^t + \beta(\nabla f_i(x^t) - v_i^t)\|^2] \right.$$

$$+ 2\beta \mathbb{E}\left[ \left\langle v_i^t - c_i^t + \beta(\nabla f_i(x^t) - v_i^t), \frac{1}{K}\sum_k \nabla f_i(y_i^{t,k}) - \nabla f_i(x^t) \right\rangle \right]$$

$$\left. + 2\beta^2 \mathbb{E}\left[ \left\| \frac{1}{K}\sum_k \nabla f_i(y_i^{t,k}) - \nabla f_i(x^t) \right\|^2 \right] + \frac{2\beta^2\sigma^2}{K} \right)$$

$$\leq \frac{q^2}{N} \sum_i \mathbb{E}\left[ \left\| v_i^t - c_i^t + \beta(\nabla f_i(x^t) - v_i^t) + \frac{\beta}{K}\sum_k (\nabla f_i(y_i^{t,k}) - \nabla f_i(x^t)) \right\|^2 \right]$$

$$+ \beta^2 q^2 L^2 U^t + \frac{2\beta^2 q^2 \sigma^2}{K}.$$

By further using Sedrakyan's inequality and Assumption 1, we obtain

$$\frac{q^2}{N} \sum_i \mathbb{E}[\|v_i^t + \beta(g_i^t - v_i^t) - c_i^t\|^2]$$

$$\leq \frac{q^2}{N} \sum_i \left( \frac{1}{q} \mathbb{E}[\|v_i^t - c_i^t\|^2] + \frac{2\beta^2}{1-q} \mathbb{E}[\|\nabla f_i(x^t) - v_i^t\|^2] \right.$$

$$\left. + \frac{2\beta^2}{1-q} \mathbb{E}\left[ \left\| \frac{1}{K}\sum_k \nabla f_i(y_i^{t,k}) - \nabla f_i(x^t) \right\|^2 \right] \right) + \beta^2 q^2 L^2 U^t + \frac{2\beta^2 q^2 \sigma^2}{K}$$

$$\leq \frac{q^2}{N} \sum_i \left( \frac{1}{q} \mathbb{E}[\|v_i^t - c_i^t\|^2] + \frac{4\beta^2}{1-q} \mathbb{E}[\|v_i^t - \nabla f_i(x^{t-1})\|^2] + \frac{4\beta^2 L^2}{1-q} \mathbb{E}[\|x^t - x^{t-1}\|^2] \right.$$

$$\left. + \frac{2\beta^2 L^2}{1-q} \frac{1}{K} \sum_i \mathbb{E}[\|y_i^{t,k} - x^t\|^2] \right) + \beta^2 q^2 L^2 U^t + \frac{2\beta^2 q^2 \sigma^2}{K}$$

$$\leq \frac{q}{N} \sum_i \mathbb{E}[\|v_i^t - c_i^t\|^2] + \frac{4\beta^2 q^2}{1-q} \frac{1}{N} \sum_i \mathbb{E}[\|\nabla f_i(x^{t-1}) - v_i^t\|^2] + \frac{4\beta^2 L^2 q^2}{1-q} \mathbb{E}[\|x^t - x^{t-1}\|^2]$$

$$+ \left(1 + \frac{2}{1-q}\right) \beta^2 q^2 L^2 U^t + \frac{2\beta^2 q^2 \sigma^2}{K}. \tag{53}$$

By combining (53) with (52) and using $1 \leq 1/(1-q)$, we finish the proof. $\qquad\square$

**Lemma 12.** *Under Assumptions 1 and 2, it holds for any $t \geq 0$ and $\eta_l K L \leq \frac{1}{2}$ that*

$$U^t$$

$$\leq \frac{9e^2 K^2 \eta_l^2}{N} \sum_i \left( \mathbb{E}[\|c_i^t - v_i^t\|^2] + \mathbb{E}[\|v_i^t - \nabla f_i(x^{t-1})\|^2] + L^2 \mathbb{E}[\|x^t - x^{t-1}\|^2] + \mathbb{E}[\|\nabla f(x^t)\|^2] \right)$$

$$+ e^2 K \eta_l^2 \sigma^2.$$

*Proof.* When $K = 1$, $U^t = 0$ trivially for all $t \geq 0$ so we consider $K \geq 2$ below. Using Young's inequality, we have

$$\mathbb{E}[\|y_i^{t,k+1} - x^t\|^2] = \mathbb{E}[\|y_i^{t,k} - \eta_l(g_i^{t,k} - c_i^t + c^t) - x^t\|^2]$$

$$\leq \mathbb{E}[\|y_i^{t,k} - \eta_l(\nabla f(y_i^{t,k}) - c_i^t + c^t) - x^t\|^2] + \eta_l^2 \sigma^2$$

$$\leq \left(1 + \frac{1}{K-1}\right) \mathbb{E}[\|y_i^{t,k} - x^t\|^2] + K \eta_l^2 \mathbb{E}[\|\nabla f(y_i^{t,k}) - c_i^t + c^t\|^2] + \eta_l^2 \sigma^2.$$

By further using Young's inequality and Assumption 1, we obtain

$$K \eta_l^2 \mathbb{E}[\|\nabla f(y_i^{t,k}) - c_i^t + c^t\|^2]$$

$$= K \eta_l^2 \mathbb{E}[\|\nabla f(y_i^{t,k}) - \nabla f_i(x^t) - (c_i^t - \nabla f_i(x^t)) + c^t - \nabla f(x^t) + \nabla f(x^t)\|^2]$$

$$\leq 3K \eta_l^2 L^2 \mathbb{E}[\|y_i^{t,k} - x^t\|^2] + 3K \eta_l^2 \mathbb{E}[\|c_i^t - \nabla f_i(x^t) - c^t + \nabla f(x^t)\|^2] + 3K \eta_l^2 \mathbb{E}[\|\nabla f(x^t)\|^2].$$

Using Young's inequality, we have

$$\frac{3K \eta_l^2}{N} \sum_i \mathbb{E}[\|c_i^t - \nabla f_i(x^t) - c^t + \nabla f(x^t)\|^2]$$

$$\leq \frac{3K \eta_l^2}{N} \sum_i \mathbb{E}[\|c_i^t - \nabla f_i(x^t)\|^2]$$

$$\leq \frac{9K \eta_l^2}{N} \sum_i \left( \mathbb{E}[\|c_i^t - v_i^t\|^2] + \mathbb{E}[\|v_i^t - \nabla f_i(x^{t-1})\|^2] + L^2 \mathbb{E}[\|x^t - x^{t-1}\|^2] \right)$$

By combining the above inequalities together, we have

$$\frac{1}{N} \sum_i \mathbb{E}[\|y_i^{t,k+1} - x^t\|^2]$$

$$\leq \left(1 + \frac{1}{K-1} + 3K \eta_l^2 L^2\right) \frac{1}{N} \sum_i \mathbb{E}[\|y_i^{t,k} - x^t\|^2] + \eta_l^2 \sigma^2$$

$$+ \frac{9K \eta_l^2}{N} \sum_i \left( \mathbb{E}[\|c_i^t - v_i^t\|^2] + \mathbb{E}[\|v_i^t - \nabla f_i(x^{t-1})\|^2] + L^2 \mathbb{E}[\|x^t - x^{t-1}\|^2] + \mathbb{E}[\|\nabla f(x^t)\|^2] \right)$$

$$\leq \cdots \leq \sum_{\ell=0}^{k} \left(1 + \frac{1}{K-1} + 3K \eta_l^2 L^2\right)^{\ell} \left( \frac{9K \eta_l^2}{N} \sum_i \left( \mathbb{E}[\|c_i^t - v_i^t\|^2] + \mathbb{E}[\|v_i^t - \nabla f_i(x^{t-1})\|^2] \right.\right.$$

$$\left.\left. + L^2 \mathbb{E}[\|x^t - x^{t-1}\|^2] + \mathbb{E}[\|\nabla f(x^t)\|^2] \right) + \eta_l^2 \sigma^2 \right)$$

$$\leq \sum_{\ell=0}^{k} \left(1 + \frac{2}{K-1}\right)^{\ell} \left( \frac{9K \eta_l^2}{N} \sum_i \left( \mathbb{E}[\|c_i^t - v_i^t\|^2] + \mathbb{E}[\|v_i^t - \nabla f_i(x^{t-1})\|^2] \right.\right.$$

$$\left.\left. + L^2 \mathbb{E}[\|x^t - x^{t-1}\|^2] + \mathbb{E}[\|\nabla f(x^t)\|^2] \right) + \eta_l^2 \sigma^2 \right), \tag{54}$$

where we use $\eta_l KL \leq \frac{1}{2}$ so that $3K\eta_l^2 L^2 \leq \frac{1}{K-1}$ in the last inequality. Iterating and averaging (54) over $k = 0, \ldots, K-1$, we obtain

$$
\begin{aligned}
U^t \leq &\frac{1}{K} \sum_k \sum_{\ell=0}^{k-1} \left(1 + \frac{2}{K-1}\right)^\ell \left(\frac{9K\eta_l^2}{N} \sum_i \left(\mathbb{E}[\|c_i^t - v_i^t\|^2] + \mathbb{E}[\|v_i^t - \nabla f_i(x^{t-1})\|^2]\right.\right. \\
&\left.\left. + L^2 \mathbb{E}[\|x^t - x^{t-1}\|^2] + \mathbb{E}[\|\nabla f(x^t)\|^2]\right) + \eta_l^2 \sigma^2\right)
\end{aligned}
$$

$$
\begin{aligned}
\leq &\sum_{\ell=0}^{K-2} \left(1 + \frac{2}{K-1}\right)^{K-1} \left(\frac{9K\eta_l^2}{N} \sum_i \left(\mathbb{E}[\|c_i^t - v_i^t\|^2] + \mathbb{E}[\|v_i^t - \nabla f_i(x^{t-1})\|^2]\right.\right. \\
&\left.\left. + L^2 \mathbb{E}[\|x^t - x^{t-1}\|^2] + \mathbb{E}[\|\nabla f(x^t)\|^2]\right) + \eta_l^2 \sigma^2\right)
\end{aligned}
$$

$$
\begin{aligned}
\leq &e^2 K\eta_l^2 \sigma^2 + \frac{9e^2 K^2 \eta_l^2}{N} \sum_i \left(\mathbb{E}[\|c_i^t - v_i^t\|^2] + \mathbb{E}[\|v_i^t - \nabla f_i(x^{t-1})\|^2]\right. \\
&\left. + L^2 \mathbb{E}[\|x^t - x^{t-1}\|^2] + \mathbb{E}[\|\nabla f(x^t)\|^2]\right)
\end{aligned}
$$

where we use the fact $\left(1 + \frac{2}{K-1}\right)^2 \leq e^2$ in the last inequality. $\square$

**Theorem 4.** *Under Assumptions 1 and 2, supposing clients are associated with $q^2$-contractive compressors, if we initialize $c_i^0 = v_i^0 = \frac{1}{B} \sum_{b=1}^B \nabla F(x^0; \xi_i^b)$, $c^0 = \frac{1}{N} \sum_{i=1}^N c_i^0$ with $\{\xi_i^b\}_{b=1}^B \overset{iid}{\sim} \mathcal{D}_i$ and $B \gtrsim \frac{\sigma^2}{(1-q)L\Delta}$ ($c_i^0 \to \nabla f_i(x^0)$ as $B \to \infty$), set*

$$
\begin{aligned}
\eta_l KL \leq &\sqrt{\frac{\beta(1-q)^2}{36e^2 N(189(1-q)^2 + 306\beta^2)}}, \quad \eta_g \eta_l KL = \left(\frac{20N}{\beta S} + \frac{28N}{(1-q)S}\right)^{-1}, \\
\beta = &\left(1 + \left(\frac{TS\sigma^2}{N^2 KL\Delta}\right)^{1/2} + \left(\frac{TS\sigma^2}{NK(1-q)L\Delta}\right)^{1/3} + \left(\frac{TS\sigma^2}{NK(1-q)^2 L\Delta}\right)^{1/4}\right)^{-1},
\end{aligned}
\tag{55}
$$

*then SCAFCOM converges as*

$$
\frac{1}{T} \sum_{t=0}^{T-1} \mathbb{E}[\|\nabla f(x^t)\|^2] \lesssim \sqrt{\frac{L\Delta\sigma^2}{SKT}} + \left(\frac{N^2 L^2 \Delta^2 \sigma^2}{(1-q)S^2 KT^2}\right)^{1/3} + \left(\frac{N^3 L^3 \Delta^3 \sigma^2}{(1-q)^2 S^3 KT^3}\right)^{1/4} + \frac{NL\Delta}{(1-q)ST}
$$

*where $\Delta \triangleq f(x^0) - \min_x f(x)$.*

*Proof.* Adding (44) $\times \frac{8\gamma N}{\beta S}$ + (51) $\times \frac{13\gamma N}{(1-q)S}$ to (37), we have

$$
\mathbb{E}[f(x^{t+1})] + \frac{8\gamma N}{\beta S} \mathbb{E}[\|v^{t+1} - \nabla f(x^t)\|^2] + \frac{14\gamma N}{(1-q)S} \frac{1}{N} \sum_i \mathbb{E}[\|v_i^{t+1} - c_i^{t+1}\|^2]
$$

$$
\begin{aligned}
\leq &\mathbb{E}[f(x^t)] + \frac{8\gamma N}{\beta S} \mathbb{E}[\|v^t - \nabla f(x^{t-1})\|^2] + \gamma\left(\frac{13N}{(1-q)S} - 1\right) \frac{1}{N} \sum_i \mathbb{E}[\|v_i^t - c_i^t\|^2] \\
&- \frac{\gamma}{2} \mathbb{E}[\|\nabla f(x^t)\|^2] - \left(\frac{1}{2\gamma} - \frac{L}{2}\right) \mathbb{E}[\|x^{t+1} - x^t\|^2] \\
&+ \gamma L^2 \left(16 + \frac{48N^2}{\beta^2 S^2} + \frac{52\beta^2 q^2}{(1-q)^2}\right) \mathbb{E}[\|x^t - x^{t-1}\|^2] \\
&+ \gamma\left(8\beta^2 + \frac{48\beta}{N} + \frac{26\beta^2 q^2}{1-q}\right) \frac{\sigma^2}{K} + \gamma L^2 \left(9\beta^2 + 48 + \frac{39\beta^2 q^2}{(1-q)^2}\right) U^t \\
&+ \gamma\left(12\beta^2 + \frac{32\beta}{N} + \frac{52\beta^2 q^2}{(1-q)^2}\right) \frac{1}{N} \sum_i \mathbb{E}[\|v_i^t - \nabla f_i(x^{t-1})\|^2].
\end{aligned}
$$

Using $q, \beta \in [0, 1]$ and $1 \leq S \leq N$ to simplify coefficients, we obtain

$$\mathbb{E}[f(x^{t+1})] + \frac{8\gamma N}{\beta S} \mathbb{E}[\|v^{t+1} - \nabla f(x^t)\|^2] + \frac{13\gamma N}{(1-q)S} \frac{1}{N} \sum_i \mathbb{E}[\|v_i^{t+1} - c_i^{t+1}\|^2]$$

$$\leq \mathbb{E}[f(x^t)] + \frac{8\gamma N}{\beta S} \mathbb{E}[\|v^t - \nabla f(x^{t-1})\|^2] + \gamma \left( \frac{13N}{(1-q)S} - 1 \right) \frac{1}{N} \sum_i \mathbb{E}[\|v_i^t - c_i^t\|^2]$$

$$- \frac{\gamma}{2} \mathbb{E}[\|\nabla f(x^t)\|^2] - \left( \frac{1}{2\gamma} - \frac{L}{2} \right) \mathbb{E}[\|x^{t+1} - x^t\|^2]$$

$$+ \gamma L^2 \left( \frac{64N^2}{\beta^2 S^2} + \frac{52\beta^2}{(1-q)^2} \right) \mathbb{E}[\|x^t - x^{t-1}\|^2]$$

$$+ \gamma \left( \frac{48\beta}{N} + \frac{26\beta^2}{1-q} \right) \frac{\sigma^2}{K} + \gamma L^2 \left( 48 + \frac{39\beta^2}{(1-q)^2} \right) U^t$$

$$+ \gamma \left( \frac{32\beta}{N} + \frac{64\beta^2}{(1-q)^2} \right) \frac{1}{N} \sum_i \mathbb{E}[\|v_i^t - \nabla f_i(x^{t-1})\|^2]. \tag{56}$$

Now adding (50) $\times 66\gamma(\frac{1}{S} + \frac{2\beta N}{(1-q)^2 S})$ to (56) and defining the Lyapunov function ($x^{-1} := x^0$)

$$\Psi^t := \mathbb{E}[f(x^t)] + \frac{8\gamma N}{\beta S} \mathbb{E}[\|v^t - \nabla f(x^{t-1})\|^2]$$

$$+ \frac{13\gamma N}{(1-q)S} \frac{1}{N} \sum_i \mathbb{E}[\|v_i^t - c_i^t\|^2] + 66\gamma \left( \frac{1}{S} + \frac{2\beta N}{(1-q)^2 S} \right) \frac{1}{N} \sum_i \mathbb{E}[\|v_i^t - \nabla f_i(x^{t-1})\|^2],$$

we obtain

$$\Psi^{t+1} - \Psi^t$$

$$\leq -\gamma \left( \frac{1}{N} \sum_i \mathbb{E}[\|v_i^t - c_i^t\|^2] + \left( \frac{\beta}{N} + \frac{2\beta^2}{(1-q)^2} \right) \frac{1}{N} \sum_i \mathbb{E}[\|v_i^t - \nabla f_i(x^{t-1})\|^2] \right)$$

$$- \frac{\gamma}{2} \mathbb{E}[\|\nabla f(x^t)\|^2] - \left( \frac{1}{2\gamma} - \frac{L}{2} \right) \mathbb{E}[\|x^{t+1} - x^t\|^2]$$

$$+ \gamma L^2 \left( \frac{64N^2}{\beta^2 S^2} + \frac{52\beta^2}{(1-q)^2} + 132 \left( \frac{N}{\beta S^2} + \frac{2N^2}{(1-q)^2 S^2} \right) \right) \mathbb{E}[\|x^t - x^{t-1}\|^2]$$

$$+ \gamma \left( \frac{32\beta}{N} + \frac{64\beta^2}{1-q} + 132 \left( \frac{\beta^2}{N} + \frac{2\beta^3}{(1-q)^2} \right) \right) \frac{\sigma^2}{K}$$

$$+ \gamma L^2 \left( 48 + \frac{39\beta^2}{(1-q)^2} + 132 \left( \frac{\beta}{N} + \frac{2\beta^2}{(1-q)^2} \right) \right) U^t.$$

Using $q, \beta \in [0, 1]$ and $1 \leq S \leq N$ to simplify coefficients, we obtain

$$\Psi^{t+1} - \Psi^t$$

$$\leq -\gamma \left( \frac{1}{N} \sum_i \mathbb{E}[\|v_i^t - c_i^t\|^2] + \left( \frac{\beta}{N} + \frac{2\beta^2}{(1-q)^2} \right) \frac{1}{N} \sum_i \mathbb{E}[\|v_i^t - \nabla f_i(x^{t-1})\|^2] \right)$$

$$- \frac{\gamma}{2} \mathbb{E}[\|\nabla f(x^t)\|^2] - \left( \frac{1}{2\gamma} - \frac{L}{2} \right) \mathbb{E}[\|x^{t+1} - x^t\|^2]$$

$$+ \gamma L^2 \left( \frac{196N^2}{\beta^2 S^2} + \frac{52\beta^2}{(1-q)^2} + \frac{264N^2}{(1-q)^2 S^2} \right) \mathbb{E}[\|x^t - x^{t-1}\|^2]$$

$$+ \gamma \left( \frac{164\beta}{N} + \frac{64\beta^2}{1-q} + \frac{264\beta^3}{(1-q)^2} \right) \frac{\sigma^2}{K} + \gamma L^2 \left( 180 + \frac{303\beta^2}{(1-q)^2} \right) U^t. \tag{57}$$

Using $9e^2K^2\eta_l^2L^2\left(180 + \frac{303\beta^2}{(1-q)^2}\right) \le \frac{\beta}{4N}$ and Lemma 12, we have

$$\gamma L^2\left(180 + \frac{303\beta^2}{(1-q)^2}\right)U^t$$

$$\le \gamma\left(\frac{1}{N}\sum_i \mathbb{E}[\|c_i^t - v_i^t\|^2] + \left(\frac{\beta}{N} + \frac{2\beta^2}{(1-q)^2}\right)\frac{1}{N}\sum_i \mathbb{E}[\|v_i^t - \nabla f_i(x^{t-1})\|^2]\right)$$

$$+ \frac{\gamma}{4}\mathbb{E}[\|\nabla f(x^t)\|^2] + \frac{\gamma L^2}{4}\mathbb{E}[\|x^t - x^{t-1}\|^2] + \frac{\gamma\beta\sigma^2}{NK}. \tag{58}$$

Plugging (58) into (57), we reach

$$\Psi^{t+1} - \Psi^t$$

$$\le -\frac{\gamma}{4}\mathbb{E}[\|\nabla f(x^t)\|^2]$$

$$- \left(\frac{1}{2\gamma} - \frac{L}{2}\right)\mathbb{E}[\|x^{t+1} - x^t\|^2] + \gamma L^2\left(\frac{197N^2}{\beta^2 S^2} + \frac{316N^2}{(1-q)^2 S^2}\right)\mathbb{E}[\|x^t - x^{t-1}\|^2]$$

$$+ \gamma\left(\frac{165\beta}{N} + \frac{64\beta^2}{1-q} + \frac{264\beta^3}{(1-q)^2}\right)\frac{\sigma^2}{K}. \tag{59}$$

Due to the choice of $\gamma = \eta_g\eta_l K$, it holds that

$$\frac{1}{2\gamma} \ge \frac{L}{2} + \gamma L^2\left(\frac{197N^2}{\beta^2 S^2} + \frac{264\beta^3}{(1-q)^2}\right),$$

Averaging (59) over $k$ and noting $\|x^0 - x^{-1}\|^2 = 0$, we obtain

$$\frac{1}{T}\sum_{t=0}^{T-1}\mathbb{E}[\|\nabla f(x^t)\|^2] \lesssim \frac{\Psi^0 - \Psi^T}{\gamma T} + \left(\frac{\beta}{N} + \frac{\beta^2}{1-q} + \frac{\beta^3}{(1-q)^2}\right)\frac{\sigma^2}{K}.$$

Note that, by the definition of $\Psi^t$, it holds that

$$\frac{\Psi^0 - \Psi^T}{\gamma T}$$

$$\lesssim \frac{L\Delta}{\gamma T} + \frac{N}{\beta S}\frac{\mathbb{E}[\|v^0 - \nabla f(x^0)\|^2]}{T}$$

$$+ \frac{N}{(1-q)S}\frac{\frac{1}{N}\sum_i\mathbb{E}[\|v_i^0 - c_i^0\|^2]}{T} + \left(\frac{1}{S} + \frac{\beta N}{(1-q)^2 S}\right)\frac{\frac{1}{N}\sum_i\mathbb{E}[\|v_i^0 - \nabla f_i(x^0)\|^2]}{T}$$

$$\lesssim \frac{L\Delta}{T}\left(\frac{N}{\beta S} + \frac{N}{(1-q)S}\right) + \frac{\sigma^2}{BT}\left(\frac{1}{\beta S} + \frac{N}{(1-q)S} + \frac{\beta N}{(1-q)^2 S}\right)$$

where we use the choice of $\gamma$ and the initialization of $\{v_i^0\}_{i\in[N]}$, $\{c_i^0\}_{i\in[N]}$, and $c^0$ in the second inequality. Due to the choice of $B$, we have

$$\frac{\sigma^2}{B}\left(\frac{1}{\beta S} + \frac{N}{(1-q)S} + \frac{\beta N}{(1-q)^2 S}\right) \lesssim L\Delta\left(\frac{N}{\beta S} + \frac{N}{(1-q)S}\right)$$

and consequently

$$\frac{1}{T}\sum_{t=0}^{T-1}\mathbb{E}[\|\nabla f(x^t)\|^2] \lesssim \frac{L\Delta}{T}\left(\frac{N}{\beta S} + \frac{N}{(1-q)S}\right) + \left(\frac{\beta}{N} + \frac{\beta^2}{1-q} + \frac{\beta^3}{(1-q)^2}\right)\frac{\sigma^2}{K}. \tag{60}$$

Plugging the choice of $\beta$ into (60) completes the proof.

$\square$

## G   IMPLEMENTATION DETAILS & MORE EXPERIMENTS

### G.1   DATASETS, ALGORITHMS AND TRAINING SETUP

**Datasets and model.** We test our algorithms on two standard FL datasets: MNIST dataset (LeCun, 1998) and Fashion MNIST dataset (Xiao et al., 2017). The MNIST dataset contains 60,000 training images and 10,000 test images. Each image is a gray-scale handwritten digit from 0 to 9 (10 classes in total) with 784 pixels. The FMNIST dataset has the same dataset sizes and the number of pixels per image whereas each image falls into 10 categories of fashion products (*e.g.*, bag, dress), making the learning task more challenging. Following (Karimireddy et al., 2020b), we train a (non-convex) fully-connected neural network with 2 hidden layers with 256 and 128 neurons, respectively. We use ReLU as the activation function and the cross-entropy loss as the training objective.

**Algorithms.** We implement our two proposed methods and two recent compressed FL algorithms, with biased and unbiased compression, respectively:

- (Biased) FED-EF (Li & Li, 2023): Federated learning with biased compression and standard error feedback. Since our proposed algorithms conduct SGD-type updates in the server, we compare them with its FED-EF-SGD variant.

- (Biased) SCAFCOM (our Algorithm 2): Biased compression for FL with stochastic controlled averaging and local momentum. The momentum $\beta$ in Algorithm 2 is tuned over a fine grid on $[0.05, 1]$.

- (Unbiased) FEDCOMGATE (Haddadpour et al., 2021): Federated learning with unbiased compression. It uses the gradient-tracking technique to alleviate data heterogeneity.

- (Unbiased) SCALLION (our Algorithm 1): Unbiased compression for FL with stochastic controlled averaging. The local scaling factor $\alpha$ is tuned over a fine grid on $[0.05, 1]$.

Besides the compressed FL algorithms, we also test the corresponding full-precision baselines: FED-SGD (also known as FEDAVG (Yang et al., 2021)) and SCAFFOLD (Karimireddy et al., 2020b), both with two-sided (global and local) learning rates. For a fair comparison, we execute SCAF-FOLD with our new implementation in experiments, corresponding to the special cases of SCAF-COM ($\mathcal{C}_i = I$, $\beta = 1$) and of SCALLION ($\mathcal{C}_i = I$, $\alpha = 1$). Notably, under a fixed random seed, our implementation yields the same training trajectory as Karimireddy et al. (2020b) at a halved uplink communication cost (by only sending one variable per participating client).

In the experiments, biased compression is simulated with TOP-$r$ operators (our Example 3). Specifically, we experiment with TOP-0.01 and TOP-0.05, where only the largest 1% and 5% entries in absolute values are transmitted in communication. For unbiased compression, we utilize random dithering (our Example 2), with 2 bits and 4 bits per entry, respectively. We tune the combination of the global learning rate $\eta_g$ and the local learning rate $\eta_l$ over the 2D grid $\{0.001, 0.003, 0.01, 0.03, 0.1, 0.3, 1, 3, 10\}^2$. The combination of learning rates with the highest test accuracy is reported for each algorithm and hyper-parameter choice (*e.g.*, $\beta$, $\alpha$, and degree of compression).

**Federated learning setting.** In our experiments, the training data are distributed across $N = 200$ clients, in a highly heterogeneous setting following (Li & Li, 2023). The training data samples are split into 400 shards each containing samples from only one class. Then, each client is randomly assigned two shards of data. Therefore, every client only possesses training samples from at most two classes. All the clients share the same initial model at $T = 0$. In each round of client-server interaction, we uniformly randomly pick $S = 20$ clients to participate in FL training, *i.e.*, the partial participation rate is 10%. Each participating client performs $K = 10$ local training steps using the local data, with a mini-batch size 32. All the presented results are averaged over 5 independent runs with the same model initialization for all the algorithms.

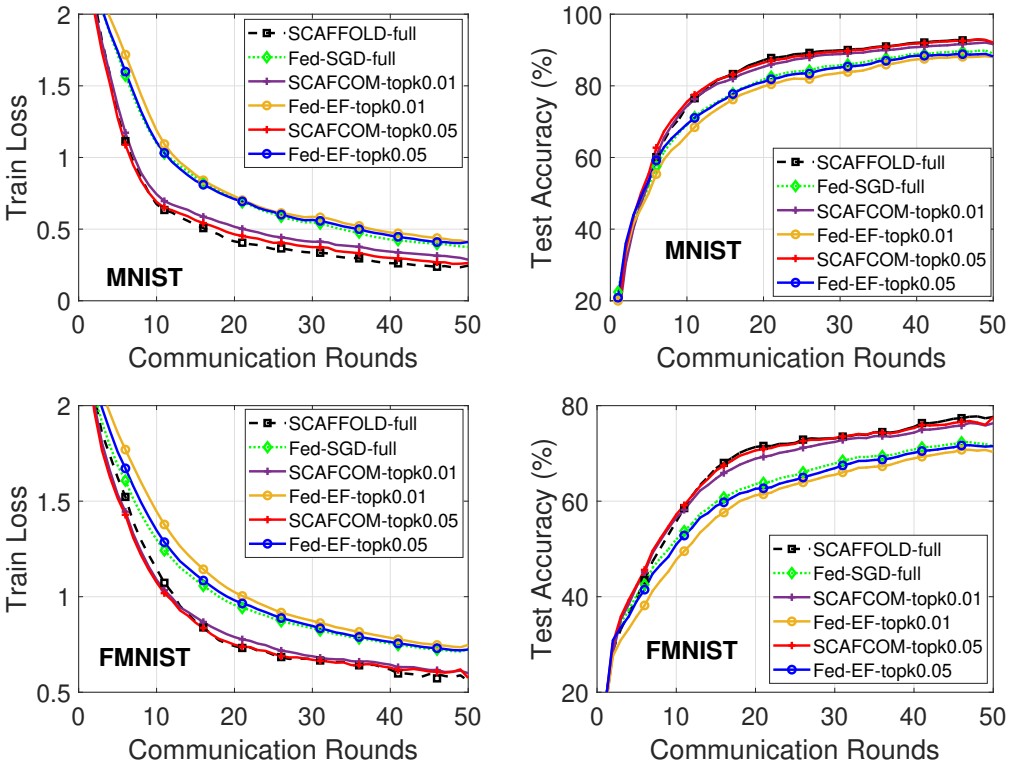

Figure 4: Train loss and test accuracy of SCAFCOM (Algorithm 2) and FED-EF (Li & Li, 2023) with biased TOP-$r$ compressors on MNIST (top row) and FMNIST (bottom row).

## G.2 RESULTS

**SCAFCOM with biased compression.** In Figure 4, we first present the train loss and test accuracy of our proposed SCAFCOM (Algorithm 2) with momentum $\beta = 0.2$ and FED-EF (Li & Li, 2023), both using biased TOP-$r$ compressors. We observe:

- In general, under the same degree of compression (*i.e.*, the value of $r$ in the case), SCAF-COM outperforms FED-EF in terms of both training loss and test accuracy, thanks to controlled variables and the local momentum in SCAFCOM.

- On both datasets, SCAFCOM with TOP-0.01 can achieve very close test accuracy as the full-precision SCAFFOLD, and SCAFCOM with TOP-0.05 essentially match those of full-precision SCAFFOLD. Hence, we can reach the same performance while saving 20 - 100x uplink communication costs.

- For both SCAFCOM and FED-EF, as the degree of compression decreases (*i.e.*, $r$ increases), their performance approaches that of the corresponding FL methods under full-precision communication (*i.e.*, SCAFFOLD and FED-SGD).

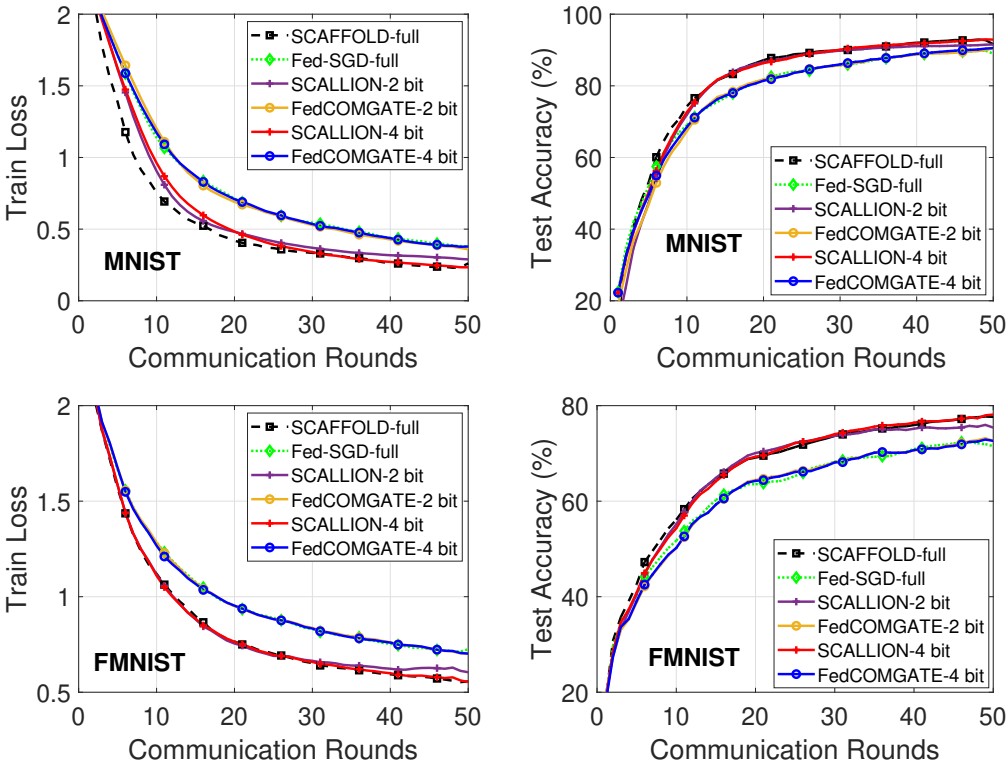

Figure 5: Train loss and test accuracy of SCALLION (Algorithm 1) and FEDCOMGATE (Haddadpour et al., 2021) with unbiased random dithering on MNIST (top row) and FMNIST (bottom row).

**SCALLION with unbiased compression.** In Figure 5, we plot the same set of experimental results and compare SCALLION ($\alpha = 0.1$) with FedCOMGATE (Haddadpour et al., 2021), both applying unbiased random dithering (Alistarh et al., 2017) with 2 and 4 bits per entry. Similarly, we see that SCALLION outperforms FedCOMGATE under the same degree of compression (number of bits per entry). The SCALLION curves of both 2-bit and 4-bit compression basically overlap that of SCAFFOLD, and 4-bit compression slightly performs better than 2-bit compression in later training rounds. Since random dithering also introduces sparsity in compressed variables, the 4-bit compressor already provides around 100x communication compression, and the 2-bit compressor saves more communication costs.

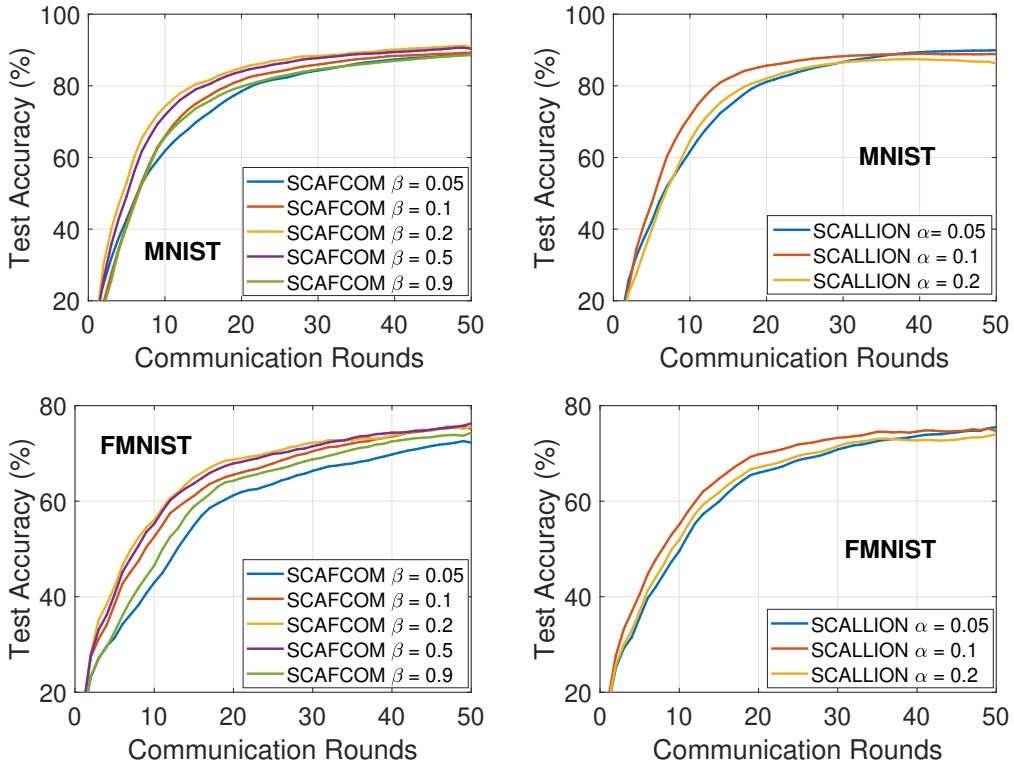

Figure 6: Test accuracy on MNIST and FMNIST of SCAFCOM (biased Top-0.01) and SCALLION (unbiased 2-bit random dithering), with different $\beta$ and $\alpha$ values.

**Impact of $\beta$ and $\alpha$.** The momentum factor $\beta$ in SCAFCOM and the scaling factor $\alpha$ in SCALLION are two important tuning parameters of our proposed methods. As an example, in Figure 6, we report the test accuracy of SCAFCOM with Top-0.01 (left column) and SCALLION (right column) 2-bit random dithering, for various $\beta$ and $\alpha$ values, respectively. From the results, we see that SCAFCOM can converge with a wide range of $\beta \in [0.05, 1]$, and $\beta = 0.2$ performs the best on both datasets (so we presented the results with $\beta = 0.2$ in Figure 1). For SCALLION, we report three $\alpha$-values, $\alpha = 0.05, 0.1, 0.2$. When $\alpha > 0.5$, the training of SCALLION becomes unstable for 2-bit quantization. As we use more bits, larger $\alpha$ could be allowed. This is because, random dithering may hugely scale up the transmitted (compressed) entries, especially for low-bit quantization. When the scaling factor $\alpha$ is too large in this case, the updates of local control variables become unstable, which further incapacitates the proper dynamic of the local/global training. Thus, for SCALLION with low-bit random dithering, we typically need a relatively small $\alpha$. As presented in Figure 2, $\alpha = 0.1$ yields the best overall performance. In general, we should tune parameters $\beta$ and $\alpha$ in SCAFCOM and SCALLION practically to reach the best performance.

