# OpenReview forum: "Stochastic Controlled Averaging for Federated Learning with Communication Compression"
_ICLR.cc/2024/Conference — ICLR 2024 spotlight_

### Official Review · Reviewer_fdMe · 2023-10-31

**Soundness:** 3 good
**Presentation:** 4 excellent
**Contribution:** 4 excellent
**Rating:** 8
**Confidence:** 4

**Summary:**

The paper delves into the challenge of minimizing the objective, which is defined as a finite sum of smooth, and potentially non-convex, functions within a Federated Learning setting. The primary focus is addressing the significant workers-to-server communication costs arising in a centralized distributed framework, which involves one main server and multiple nodes. The authors' contributions can be summarized as:

- They present a novel formulation of the foundational SCAFFOLD algorithm, which effectively cuts uplink communication expenses by half. This lays a more straightforward foundation for integrating communication compression.
- The SCALLION algorithm is introduced, leveraging the new SCAFFOLD formulation combined with unbiased compressors.
- The SCAFCOM algorithm is developed to facilitate biased compressors in Federated Learning via local momentum. Convergence analysis for standard contractive compressors is provided.
- It is demonstrated that both SCALLION and SCAFCOM either match or surpass the communication and computation complexities of current compressed FL baselines.
- Through experiments, it's evident that SCALLION and SCAFCOM deliver performance akin to full-precision techniques, boasting compression savings exceeding 100x.

**Strengths:**

- The writing of the paper is clear, the main claims are outlined, and the text is easily read;
- Literature review is solid and contains relevant papers on the topic;
- Proposed novel FL algorithms, SCALLION and SCAFCOM, are provably shown to be robust to heterogeneous data and partial participation and require only standard assumptions to establish first-order stationary point guarantees;
- SCALLION attains superior convergence guarantees compared to prior compressed FL methods with unbiased compression under minimal assumptions;
- Local momentum in SCAFCOM overcomes the adverse effects of biased compression. SCAFCOM improves communication complexity by a factor of 1/(1-q) over the prior art;
- Both SCALLION and SCAFCOM exhibit robustness to heterogeneous data and partial participation, unlike existing approaches;
- The paper provides a principled way to integrate communication compression into federated learning through the new SCAFFOLD formulation;
- Algorithms are simple to implement and empirically achieve significant compression savings.

**Weaknesses:**

1) Some potentially relevant papers are missing in the references. See "Suggestions" below.

## Typos:
1) In the RELATED WORK section, instead of "TurnGrad" it should be "TernGrad".

## Suggestions:
I would recommend authors to reconsider some of the phrases that they wrote, in particular

>>Can we design FL approaches that accommodate arbitrary data heterogeneity, local updates, and partial participation, as well as support communication compression?
>>
>>In the literature, none of the existing algorithms have successfully achieved this goal, to the best of our knowledge.

To the best of my knowledge, this is true, but for the general class of non-convex loss functions. However, this is not true for strongly convex functions. There are already some works [1, 2, 3, 4] on that topic, providing a provably beneficial combination of compression, partial participation, and local updates.

[1] Grudzień, M., Malinovsky, G., & Richtárik, P. (2023). Improving Accelerated Federated Learning with Compression and Importance Sampling. _arXiv preprint arXiv:2306.03240_.

[2] Youn, Y., Kumar, B., & Abernethy, J. (2022, October). Accelerated Federated Optimization with Quantization. In _Workshop on Federated Learning: Recent Advances and New Challenges (in Conjunction with NeurIPS 2022)_.

[3] Condat, L., Malinovsky, G., & Richtárik, P. (2023). TAMUNA: Accelerated federated learning with local training and partial participation. _arXiv preprint arXiv:2302.09832_.

[4] Sadiev, A., Malinovsky, G., Gorbunov, E.A., Sokolov, I., Khaled, A., Burlachenko, K., & Richt'arik, P. (2022). Federated Optimization Algorithms with Random Reshuffling and Gradient Compression. _ArXiv, abs/2206.07021_.

**Questions:**

There are several questions related to the prof of the Theorem 3:
1) (Page 25) Could authors please clarify why this inequality holds?

$$9 e^2 K^2 \eta_l^2 L^2\left(24+\frac{4(1+\omega) \alpha^2}{S}+\frac{522(1+\omega) \alpha}{N}\right) \leq \frac{\alpha(1+\omega)}{N} \leq \frac{1}{4}$$

2) Why does this property hold?
$$\| {x^0 - x^1}\|=0$$

## Conclusion
I would happily give this paper a higher grade for its theoretical contributions and reliable experiments. However, at this moment, I can not do so since the paper still contains things that need to be clarified. I am ready to reconsider my current rate during rebuttals once you respond to me on Weaknesses and Questions.

## Update after the author's rebuttal
I increased the score.

---

> ### Author Response · Authors · 2023-11-16
>
> Dear Reviewer fdME:
>
> Thanks for your valuable feedback. All questions have been clarified as best as we can and all the revisions in the updated manuscript are highlighted with the red color.
>
> - **Softening the claim.** Thanks for the suggestion and for mentioning the related papers. We have modified the claim in our updated manuscript with the additional references by saying ``In the literature, none of the existing algorithms have successfully achieved this goal in non-convex
> FL, despite a few studies in the strongly convex scenarios (Grudzie ́n et al., 2023; Youn et al., 2022;
> Condat et al., 2023; Sadiev et al., 2022)''. We hope this addresses your question.
>
> - **Clarification for the inequality.** We highly appreciate your efforts in carefully reading through the proofs. There is indeed a miscalculation here in our earlier manuscript. We have fixed it and highlighted the revision with the red color on page 25. Note that this minor miscalculation does not affect the soundness of our theories as we essentially only need $\eta_l^2 K^2L^2 \lesssim \alpha (1+\omega)/N$. The existence of numerical numbers does not affect the final convergence rate.
>
> - **Clarification for $x^{-1}=x^{0}$.**
> We use $x^{-1}:=x^{0}$ notationally in our proofs instead of $x^1=x^{0}$. We think reviewer fdME meant to ask the clarification regarding $x^{-1}=x^{0}$.
>  The variable $x^{-1}:=x^0$ is introduced to simplify the notations in telescoping and does not appear in our algorithmic iterations which indeed start with $t=0$. Specifically, our proof frequently uses the term $\mathbb{E}[\\|x^t-x^{t-1}\\|^2]$, which typically emerges from breaking $\mathbb{E}[\\|c_i^t-\nabla f_i(x^{t})\\|^2]$ and $\mathbb{E}[\\|c^t-\nabla f(x^{t})\\|^2]$ for $t\geq 1$ as
> \begin{equation}\mathbb{E}[\\|c_i^t-\nabla f_i(x^{t})\\|^2]\lesssim \mathbb{E}[\\|c_i^t-\nabla f_i(x^{t-1})\\|^2]+L^2\mathbb{E}[\\|x^t-x^{t-1}\\|^2]\end{equation}
> and
> $$\mathbb{E}[\\|c^t-\nabla f(x^{t})\\|^2]\lesssim \mathbb{E}[\\|c^t-\nabla f(x^{t-1})\\|^2]+L^2\mathbb{E}[\\|x^t-x^{t-1}\\|^2],$$
> see, e.g., Lemma 4, 5, 6.
> By hypothetically letting $x^{-1}=x^0$, the above inequalities hold trivially for $t=0$. We hope this notation is clear now.
>
> Thanks again for your feedback on our work.

---

> > ### Author Response · Authors · 2023-11-20
> >
> > Dear Reviewer fdMe,
> >
> > We hope our rebuttal has answered your questions and clarified the concerns. Please kindly let us know if there are additional questions we should address, before the interactive rebuttal system is closed.
> >
> > Thank you and happy thanksgiving.

---

> > > ### Comment · Reviewer_fdMe · 2023-11-21
> > > **Score is increased; further recommendations on literature review improvement are provided**
> > >
> > > Dear Authors,
> > >
> > > Thank you for addressing my previous concerns and queries in your latest revisions. I am pleased to see that the issues raised have been thoughtfully considered and rectified. The additional clarifications and adjustments you have made to the manuscript enhance its quality and comprehensiveness.
> > >
> > > I have reassessed my evaluation of your submission and have adjusted my grade accordingly to reflect the improvements made. The paper now more effectively situates itself within the broader context of federated learning research, and the amendments to the theoretical aspects are particularly commendable.
> > >
> > > Additionally, I would like to bring to your attention three more partially related articles that could further enrich the context of your work, especially in the realm of non-convex federated learning. The first paper by Richtárik et al. generalizes the framework proposed in EF21, and the second one by Fatkhullin et al. extends this framework to encompass various scenarios including stochastic updates and partial participation, albeit without employing local steps.
> > > Finnaly, the work by Zhao et al. claims to improve a non-convex rate of Scaffold by a certain factor. Although this paper does not incorporate a compression mechanism, its mention in your literature review could provide a more comprehensive overview of advancements in this domain.
> > >
> > > Thank you again for your diligent work and responsiveness to the review process. I look forward to the potential incorporation of these suggestions in your final manuscript.
> > >
> > > ### References:
> > >
> > > [1] Richtárik, et al. “3PC: Three point compressors for communication-efficient distributed training and a better theory for lazy aggregation.” 39th International Conference on Machine Learning (ICML), 2022.
> > >
> > > [2] Fatkhullin, et al. "EF21 with bells & whistles: Practical algorithmic extensions of modern error feedback." arXiv preprint arXiv:2110.03294 (2021).
> > >
> > > [3] Zhao et al. "FedPAGE: A fast local stochastic gradient method for communication-efficient federated learning." arXiv preprint arXiv:2108.04755 (2021).

---

> > > > ### Author Response · Authors · 2023-11-21
> > > > **Thank you**
> > > >
> > > > Dear Reviewer fdMe
> > > >
> > > > Thanks so much for the thoughtful reassessing of our work. We will carefully study the additional reference you kindly suggested and will incorporate them into the final version if they are relevant.
> > > >
> > > > Sincerely,
> > > >
> > > > Authors

---

### Official Review · Reviewer_17jD · 2023-11-01

**Soundness:** 3 good
**Presentation:** 4 excellent
**Contribution:** 3 good
**Rating:** 8
**Confidence:** 3

**Summary:**

In this work, the authors propose two new algorithms SCALLION and SCAFCOM. Those new FL algorithms show robustness to data heterogeneity, partial participation, local updates and use communication compression. They are based on SCAFFOLD algorithm.

The authors provide convergence analysis for these two algorithms in non-convex case and show that the convergence rate is faster than rates of previous algorithms. The experiments support theoretical guarantees obtained by the authors.

**Strengths:**

1. Interesting idea related broadcasting the compressed difference. This new view on the updates from SCAFFOLD help to design new proposed algorithms and to understand why the work.
2. Only two assumptions are used for convergence analysis.
3. Well written paper and good presentation of results. It is easy to follow.

**Weaknesses:**

1. For me there is no reasonable weaknesses. Possible, in camera ready version it would be better compare your methods with this work
https://arxiv.org/pdf/2310.07983.pdf .

**Questions:**

I do not have questions. Probably, I will ask some questions during the discussion period.

Typos:
1. In the third row of the chain of inequalities, $L^2$ is missed in the second term.

**Details Of Ethics Concerns:**

-

---

> ### Author Response · Authors · 2023-11-16
>
> Dear Reviewer 17jD:
>
> Thanks for your valuable feedback and support.
>
> We have cited and discussed the paper https://arxiv.org/pdf/2310.07983.pdf in the section of related works in the updated manuscript. However, since this method does not support communication compression, we do not
> compare our methods with it directly.
>
> We tried our best to look for the typo you pointed out. However, we did not find the location. Could you please kindly inform us of the location where $L^2$ is missing?
>
> Thanks again for your feedback on our work.

---

> > ### Comment · Reviewer_17jD · 2023-11-22
> >
> > Thank you for your response!
> > > Could you please kindly inform us of the location where $L^2$  is missing?
> >  I apologize for forgetting to mention the exact page: p. 28 eq. (39).

---

> > > ### Author Response · Authors · 2023-11-22
> > >
> > > We highly appreciate your efforts in carefully reading through the proofs. We have fixed the typos and highlighted the revision with the red color in the updated manuscript. Thank you again for your valuable input.

---

### Official Review · Reviewer_aALW · 2023-11-07

**Soundness:** 3 good
**Presentation:** 3 good
**Contribution:** 3 good
**Rating:** 8
**Confidence:** 2

**Summary:**

The paper suggests a federated learning algorithm that finds a stationary point while handling arbitrary client heterogeneity, partial client participation, local updates, and gradient compression. This work introduces an algorithm based on SCAFFOLD which communicates to the server only a single compressed vector. The paper shows convergence to a stationary point under very weak communication and compression assumptions and provides results for both biased and unbiased compressors.

**Strengths:**

* The proof looks sound.
* I believe that the results for biased compressors are a substantial contribution.

**Weaknesses:**

1) I believe the paper lacks comparison (both theoretical and experimental) with "MARINA: Faster Non-Convex Distributed Learning with Compression". Both papers pursue the same goals, namely handling the following FL issues:
* arbitrary client heterogeneity,
* partial client participation,
* gradient compression.

Overall, as far as I know, MARINA (one of itsvariations) is the closest result in terms of settings (and I think it achieves similar bounds), and I'm not sure whether the authors are aware of it.

2) Theorems 1 and 2: "set learning rates $\eta_l$ and $\eta_g$ as well as scaling factor $\alpha$ properly" - the parameters should be specified in the main body.

3) The paper handles local updates (K local updates per round), but this is achieved by dividing the learning rate by a factor of K. In other words, local updates don't provide provable improvement compared to a single larger gradient step.

**Questions:**

How does your paper compare with MARINA?

---

> ### Author Response · Authors · 2023-11-16
>
> Dear Reviewer aALW:
>
> Thanks for your valuable feedback on our work. All questions have been clarified as best as we can and the revisions in the updated manuscript are highlighted with the red color.
>
> **1. Comparison with MARINA**
>
> We did not compare our algorithms with MARINA in the original manuscript for a couple of reasons. (i) MARINA does not support local updates, the fundamental characteristic of federated learning algorithms. (ii) MARINA lacks the guarantee of supporting partial client participation and stochastic gradients simultaneously. Specifically, VR-MARINA supports stochastic gradients but not partial participation while  PP-MARINA supports partial participation but not stochastic gradients. (iii) MARINA requires  the averaged smoothness (see Assumption 3.2 therein), which is beyond the scope of our manuscript. Nevertheless, we are happy to cite the paper and add some discussions.
>
> We have supplemented the complexities of MARINA in Table 1 in the updated manuscript. We observe that SCAFCOM outperforms MARINA in terms of computation complexity but is inferior in terms of communication complexity. Despite the comparison of theoretical rates, we again would like to emphasize that the problem setups are different, and SCALLION and SCAFCOM can be advantageous in the practical federated learning setup where local updates, stochastic gradients, and partial participation are entangled together.
>
>
> **2. Specifications of learning rates and scaling factors**
>
> We have specified the choices of $\eta_l$, $\eta_g$, $\alpha$, and $\beta$ in the updated manuscript (in red color).
>
>
> **3. No provable improvement compared to a single larger gradient step**
>
> To our knowledge, the shrinkage of the learning rate due to local updates is customary in non-convex and stochastic FL literature.  The main focus of this work is to address the entanglement of local updates, stochastic gradients, partial participation, compression, and data heterogeneity that appears naturally in practical FL.
>
> We remark that the provable benefit of local updates is a long-standing problem particularly in the stochastic and non-convex scenarios, without a bounded-similarity-type condition being imposed over $\\{f_i\\}_{i=1}^N$. While there is a stream of works demonstrating the benefit as mentioned by Reviewer fdMe in strongly convex scenarios, we are not aware of any existing work achieving this goal in the stochastic and non-convex scenarios. We thus intend to leave this ambitious goal for future work.
>
> Please kindly let use know if there are any further questions. Thanks again for your review of our work.

---

> > ### Comment · Reviewer_aALW · 2023-11-19
> >
> > Thank you for your replies. I updated my score.
> >
> > > Comparison with MARINA
> > Thank you for the explanation. I had PP-MARINA in mind, but I overlooked the lack of stochastic gradients.
> >
> > > We have specified the choices of [parameters]  in the updated manuscript:
> >
> > Thank you. I think it is fine to specify the parameters only up to big-O notation.
> > I think it would be great to explain where all the terms come from.
> > I also suspect that you can present a specific choice of parameters that would preserve the complexity while drastically simplifying the presentation.

---

> ### Author Response · Authors · 2023-11-19
>
> We are delighted that your concerns have been resolved, and we sincerely appreciate your positive feedback. We will incorporate your suggestions to streamline the choices of parameters in our later revisions. Thank you again for your valuable input.

---

### Meta-Review · Area_Chair_thet · 2023-12-06

**Metareview:**

This research introduce a novel federated learning algorithm aimed at handling client heterogeneity, partial participation, local updates, and gradient compression, while maintaining convergence to a stationary point.
One of the main contributions is the extension of a momentum EF21 algorithm, recently proposed by Fatkhullin et al. (NeurIPS 2023), to the federated setting. The convergence of the resulting method, denoted as SCAFCOM, does not depend on the data-heterogeneity, and improves over prior methods also under client-sampling.

This is a solid technical paper, and the reviewers unanimously find the results to be of interest to the ICLR community.

On a minor note, the efficient implementation formulation of Scaffold might be identical to the one presented in (Ye et al, NeurIPS 2022, https://arxiv.org/abs/2207.06343). The authors are encouraged to discuss the similarity between their algorithm and this existing formulation.
Please also remove the (private) author comments in the final version.

**Justification For Why Not Higher Score:**

While the contribution of this work is solid, it doesn't seem to be groundbreaking. Instead, it appears to be more of a continuation of a line of work, notably Fatkhullin et al. (NeurIPS 2023).

**Justification For Why Not Lower Score:**

All reviewer agree on acceptance, and it seems to be a first algorithm that can jointly handle client heterogeneity, partial participation, local updates, and uplink gradient compression.

---

### Decision · Program_Chairs · 2024-01-16

Accept (spotlight)